# G protein-coupled receptor 151 regulates glucose metabolism and hepatic gluconeogenesis

Ewa Bielczyk-Maczynska [1,2,3] ✉, Meng Zhao [2,3,4], Peter-James H. Zushin [5], Theresia M. Schnurr[1,2,3], Hyun-Jung Kim [1], Jiehan Li[1,2,3], Pratima Nallagatla[6], Panjamaporn Sangwung[1,2,3], Chong Y. Park[1,2,3], Cameron Cornn[1], Andreas Stahl [5], Katrin J. Svensson [2,3,4] & Joshua W. Knowles [1,2,3,7] ✉

Human genetics has been instrumental in identification of genetic variants linked to type 2 diabetes. Recently a rare, putative loss-of-function mutation in the orphan G-protein coupled receptor 151 (*GPR151*) was found to be associated with lower odds ratio for type 2 diabetes, but the mechanism behind this association has remained elusive. Here we show that *Gpr151* is a fasting- and glucagon-responsive hepatic gene which regulates hepatic gluconeogenesis. *Gpr151* ablation in mice leads to suppression of hepatic gluconeogenesis genes and reduced hepatic glucose production in response to pyruvate. Importantly, the restoration of hepatic *Gpr151* levels in the *Gpr151* knockout mice reverses the reduced hepatic glucose production. In this work, we establish a previously unknown role of *Gpr151* in the liver that provides an explanation to the lowered type 2 diabetes risk in individuals with nonsynonymous mutations in *GPR151*.

Type 2 diabetes (T2D) is a major health problem worldwide. There is a great need for deeper understanding of molecular mechanisms and identification of novel drug targets to correct abnormal glucose metabolism that characterizes T2D[1,2]. Human genetics is being increasingly used to identify potential drug targets to combat T2D and cardiovascular disease, including identifying new functions for orphan G-protein coupled receptors (GPCRs)[3,4]. GPCRs are a superfamily of over 800 highly druggable receptors and are targeted by ~34% of the current FDA-approved drugs[5].

Recently, a rare nonsynonymous, presumed inactivating, mutation (p.Arg95Ter) in the gene encoding the orphan G-protein coupled receptor 151 (*GPR151*), a $G\alpha_{o1}$-linked GPCR, was associated with lower odds ratio for T2D, obesity and coronary artery disease[6] and with reduced body-mass index (BMI)[6-8], although another recent study found no significant associations between putative loss-of-function (LOF) *GPR151* variants and BMI, T2D, or other metabolic traits[9].

The function of GPR151 was first described in the habenula, a brain structure with crucial role in processing reward-related aversive signals[10]. In mice, whole-body knockout (KO) of *Gpr151* results in diminished behavioral responses to nicotine, including less pronounced suppression of appetite[11], The nonsynonymous *GPR151* p.Arg95Ter variant is linked to 12% lower odds of clinical obesity[6] and a 14% decrease in the odds of T2D[6,7], which suggests the presence of a habenula-mediated regulation of appetite or other mechanisms which mediate the effect of *GPR151* LOF variants on decreased odds of T2D. In addition to the brain, *Gpr151* is widely expressed in peripheral tissues in mice, including metabolically relevant tissues such as brown and white adipose tissue, liver, skeletal muscle, and pancreas[12]. However, the peripheral functions of GPR151 and its mechanism of action in controlling glucose metabolism are entirely unknown.

Several mechanisms sustain glucose homeostasis in the body, including glucose production by the liver, kidney and gut, as well as

[1]Division of Cardiovascular Medicine, Department of Medicine, Stanford University School of Medicine, Stanford, CA, USA. [2]Stanford Diabetes Research Center, Stanford University School of Medicine, Stanford, CA, USA. [3]Stanford Cardiovascular Institute, Stanford University School of Medicine, Stanford, CA, USA. [4]Department of Pathology, Stanford University School of Medicine, Stanford, CA, USA. [5]Department of Nutritional Sciences and Toxicology, University of California at Berkeley, Berkeley, CA, USA. [6]Genetics Bioinformatics Service Center, Stanford University School of Medicine, Stanford, CA, USA. [7]Stanford Prevention Research Center, Stanford University School of Medicine, Stanford, CA, USA. ✉e-mail: ewabm@stanford.edu; knowlej@stanford.edu

glucose uptake by skeletal muscle, heart muscle and adipose tissue[2]. T2D is associated with increased rates of gluconeogenesis[13] and impaired glucose uptake in peripheral tissues due to insulin resistance[14,15]. In diabetic patients[16] and in mouse models[17], excessive signaling through the glucagon receptor, a $G_s$ alpha subunit-coupled GCPR, contributes to pathologically elevated hepatic gluconeogenesis. This effect is mediated through an increase in intercellular cyclic AMP (cAMP) levels[18]. Surprisingly, excessive signaling through the $G\alpha_i$ class of G proteins, which inhibits cAMP production, can also trigger an increase in hepatic gluconeogenesis in mice[19]. Moreover, a lack of functional $G\alpha_i$-type G proteins in mouse hepatocytes reduces blood glucose levels[19]. Therefore, the relationship between the dynamics of cAMP signaling in the liver and hepatic gluconeogenesis is complex. In addition, the identity and role of $G\alpha_i$-linked GPCRs which regulate hepatic gluconeogenesis are currently unknown.

Here, we dissect the mechanism of action of *Gpr151* in vivo and in vitro. In mice, *Gpr151* expression in the liver is increased by fasting. Whole-body loss of *Gpr151* confers increased glucose tolerance in high-fat diet-induced obesity. Furthermore, we show that GPR151 has a cell-autonomous role in hepatic gluconeogenesis and that loss of *Gpr151* is protective for metabolic health in diet-induced obesity through decreasing gluconeogenesis in hepatocytes. Liver-specific *Gpr151* overexpression in *Gpr151* knockout mice abrogates these positive effects resulting in increased hepatic gluconeogenesis. We query summary statistics from published genome-wide association studies (GWAS) in humans to identify associations between the *GPR151* p.Arg95Ter LOF variant and selected metabolic traits. Lastly, we confirm that the p.Arg95Ter variant leads to the absence of GPR151 protein in an overexpression study in vitro. In summary, our results demonstrate a function for GPR151 in regulating glucose metabolism.

## Results

### Gpr151 KO improves glucose metabolism in diet-induced obesity

Given the association between *GPR151* LOF variant p.Arg95Ter and lower odds ratio for T2D, obesity and reduced BMI, we examined the function of GPR151 with respect to metabolic health in diet-induced obesity (DIO). We employed the previously described whole-body *Gpr151* KO mouse model, which was used to determine the conserved expression and role of GPR151 in the habenula, a brain structure critical for processing of reward-related and aversive signals[10,11]. As the LOF *GPR151* variant was associated with lower BMI in humans, we compared body weights of *Gpr151* wild-type (WT) and knockout littermates fed either standard diet (SD) or obesity-inducing high-fat diet (HFD; Fig. 1a). Body weights of *Gpr151* KO animals did not differ significantly from WT littermates, irrespective of sex or diet (Fig. 1b, Extended Data Figs. 1 and 2).

Next, the effect of *Gpr151* KO on whole-body glucose metabolism was assessed using glucose and insulin tolerance testing. *Gpr151* KO mice showed dramatically improved glucose tolerance compared to WT littermates in DIO but not when fed standard diet (Fig. 1c and Extended Data Figs. 1 and 2). There were no significant differences in insulin tolerance between *Gpr151* WT and KO mice in DIO conditions, suggesting that GPR151 regulates blood glucose levels but not insulin action per se (Fig. 1d). While *Gpr151* KO mice showed lower fasting blood glucose levels (Fig. 1e), their plasma insulin levels did not differ from WT littermates in DIO (Fig. 1f). Because of the known role of GPR151 in the regulation of appetite[11] we conducted an analysis of food intake, activity, and metabolic parameters in *Gpr151* WT and KO DIO mice using Comprehensive Lab Animal Monitoring System (CLAMS) to identify a mechanism which could explain the improved glucose metabolism in *Gpr151* KO mice. In spite of small increases in food intake in *Gpr151* KO mice compared to WT littermates in this cohort (Extended Data Fig. 3), there were no significant differences in oxygen consumption (Fig. 1g, h), $CO_2$ production (Fig. 1I, j), or respiratory exchange ratio (Fig. 1k), indicating no differences in the usage of

carbohydrates and lipids for energy and no differences in sympathetic activity in adipose tissue. In addition, locomotion and energy expenditure in *Gpr151* KO and WT mice were comparable (Fig. 1l–n), indicating that behavioral differences are not the cause of the differences in glucose metabolism.

In summary, whole-body *Gpr151* KO resulted in improved whole-body glucose metabolism in DIO mice, which could not be explained by differences in body composition or physical activity.

### Gpr151 expression in liver is decreased by feeding

To determine which tissues may mediate the effect of GPR151 on whole-body glucose metabolism, we quantified *Gpr151* expression in a tissue panel from WT mice. Compared to the brain, known to express *Gpr151*[10], several tissues showed higher *Gpr151* expression, including skeletal muscle and liver (Fig. 2a). To identify tissues in which *Gpr151* expression is regulated by changes in the metabolic state, we compared gene expression in SD-fed and DIO WT mice, focusing on the top ten *Gpr151*-expressing tissues, as well as adipose tissue due to its role in glucose metabolism[20]. In the DIO model, *Gpr151* was significantly upregulated in the hind brain and pituitary gland and downregulated in the liver and subcutaneous white adipose tissue (Fig. 2b). Next, we focused on peripheral tissues involved in glucose metabolism (liver, skeletal muscle, brown and white adipose tissue) and compared *Gpr151* expression under fasting and feeding conditions. *Gpr151* expression was robustly downregulated in the liver following feeding (Fig. 2c). This postprandial regulation of *Gpr151* gene expression in the liver resembled that of *Fgf21* and *Pck1* (Fig. 2d), whose expression levels are tightly regulated by fasting and feeding[21].

To further determine the physiological regulation of *Gpr151*, we assessed *Gpr151* expression following the injection of insulin and glucagon, the major hormones which regulate carbohydrate metabolism. Insulin injection led to a significant downregulation of *Gpr151* expression in the white adipose tissue but not the liver (Fig. 2e). In contrast, glucagon induced a significant upregulation of *Gpr151* in the liver (Fig. 2f). These data indicate that *Gpr151* expression is regulated by insulin and glucagon in peripheral tissues involved in glucose metabolism. However, the molecular mechanisms behind the regulation of *Gpr151* by blood glucose-regulating hormones in adipose tissue and liver are distinct. Given the high hepatic expression of *Gpr151* compared with the adipose tissue, *Gpr151* in the liver is likely to be more physiologically important (Fig. 2a).

To further characterize the regulation of *Gpr151* expression in the liver, we investigated and identified transcription factor binding sites upstream of the human *GPR151* transcript for the cAMP-responsive element-binding protein (CREB) and glucocorticoid receptor (GR) which are conserved in mice (Extended Data Fig. 4a, b). The activity of CREB is regulated by glucagon, insulin[22] and glucocorticoids. Glucocorticoid blood levels increase during prolonged fasting and stimulate hepatic gluconeogenesis through GR[23]. We tested whether CREB and GR induce *Gpr151* expression in mouse hepatocyte AML12 cell line following cAMP induction by forskolin treatment, GR activation by dexamethasone treatment, or both. *Gpr151* expression increased by forskolin but not by dexamethasone (Extended Data Fig. 4c), and the induction of *Gpr151* expression by forskolin was abrogated by co-treatment with the CREB inhibitor 666-15[24] (Extended Data Fig. 4d). Gene expression induction by forskolin was unique to *Gpr151*, compared to other genes encoding $G_i$-interacting GPCRs that are known to be expressed in the liver (Extended Data Fig. 4e). We concluded that *Gpr151* expression in the liver may be at least partially upregulated by cAMP signaling through CREB. However, considering the strong upregulation of *Gpr151* expression observed in the livers of fasting mice (Fig. 2c), additional regulators of *Gpr151* expression in addition to CREB might be present.

In conclusion, we found that *Gpr151* expression is downregulated in the liver and adipose tissue by feeding, which strongly suggests that

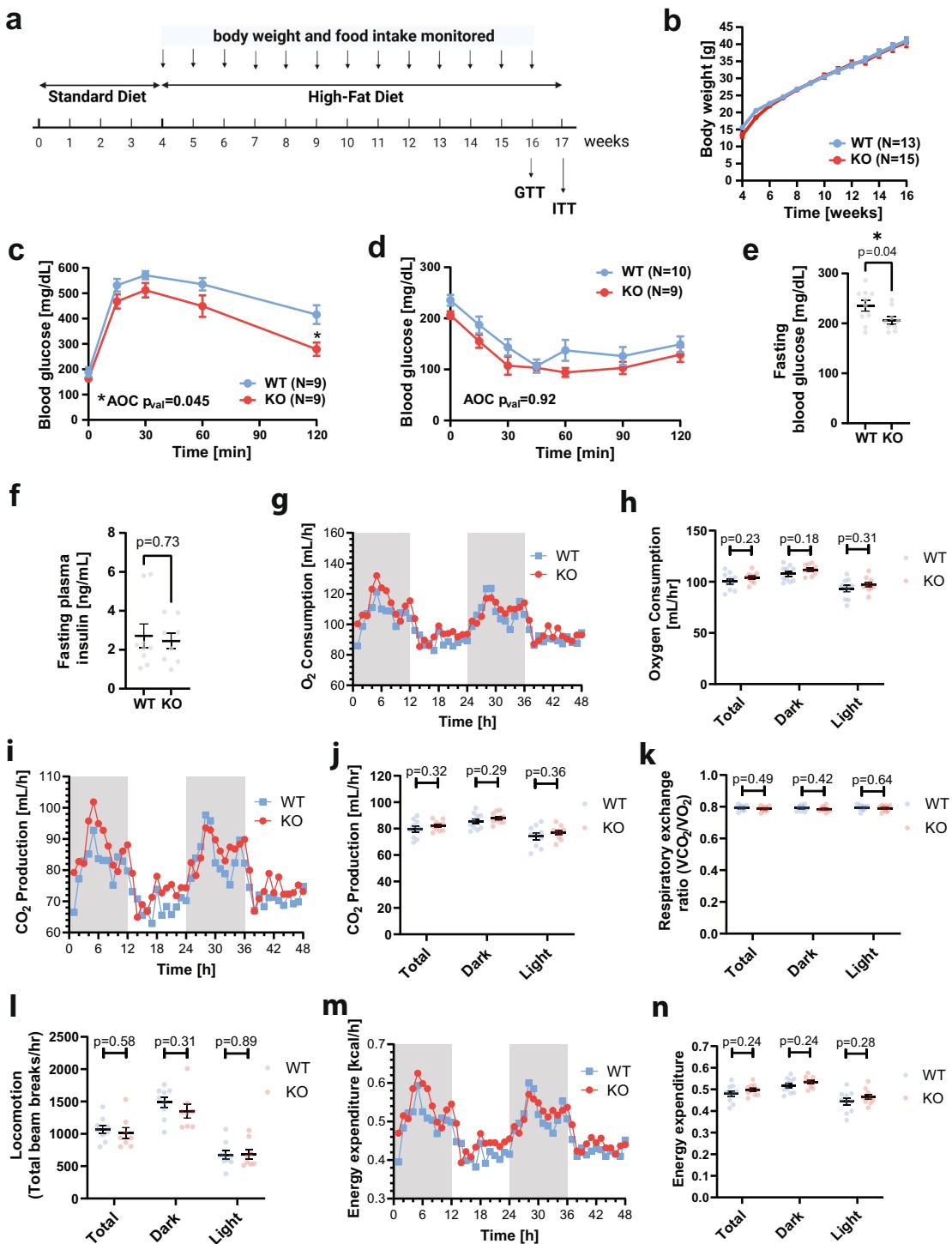

the effects of *Gpr151* ablation on whole-body glucose metabolism is mediated by these peripheral metabolic tissues.

### *Gpr151* loss impairs hepatic glucose production

To determine the mechanistic role of *Gpr151* in the liver, we verified the absence of *Gpr151* transcript in the livers of *Gpr151* KO mice using RT-qPCR (Fig. 3a and Extended Data Fig. 5). We compared the liver transcriptomes from *Gpr151* WT and KO DIO mice using bulk RNA sequencing (RNA-Seq; Fig. 3b). The analysis revealed 79 significantly upregulated ($p$-val$_{adj}$ < 0.05, log2Fold change >1) and 338 significantly downregulated ($p$-val$_{adj}$ < 0.05, log2Fold change < (−1)) genes in *Gpr151* KO livers compared to WT littermates (Fig. 3c, Source Data). To gain

biological insights from the transcriptome changes, we conducted Gene Set Enrichment Analysis (GSEA) using the Hallmark gene sets[25], which revealed significant enrichment (FDR $q$-val < 0.25) in 32 gene sets in WT livers and no gene sets significantly enriched in KO livers (Source Data). Surprisingly, the expression of genes in the glycolysis and gluconeogenesis pathway was decreased in KO livers (Fig. 3d). Downregulation of the expression of several genes from this pathway in the livers of *Gpr151* KO mice was further validated using RT-qPCR in female littermate *Gpr151* WT and KO DIO mice (Fig. 3e). Expression of *Ppargc1a*, a gene encoding transcriptional coactivator PPARGC1A which regulates the expression of genes involved in energy metabolism[26], was diminished in the livers but not in skeletal muscle or

**Fig. 1 | *Gpr151* loss improves glucose metabolism in DIO male mice which is not explained by behavioral differences. a** Schematic of the experiment to determine the effect of *Gpr151* loss on metabolic health in mice using the DIO model. Image created using Biorender. **b** Body weight in male DIO KO and WT mice over 12 weeks of HFD ($N = 13$, WT; $N = 15$, KO). Data are presented as mean values ±SEM. **c** Blood glucose levels measured during glucose tolerance testing in *Gpr151* KO and WT DIO males. Data are presented as mean values ±SEM. Area of the curve (AOC) compared using two-tailed Student's $t$ test. Student's $t$ test with Bonferroni correction used to test differences at every time point ($N = 9$, WT; $N = 9$, KO; *$t = 120$ $q_{val} = 0.02$). **d** Blood glucose levels measured during insulin tolerance testing in *Gpr151* KO and WT in DIO male mice ($N = 10$, WT; $N = 9$, KO). Data are presented as mean values ±SEM. AOC compared using two-tailed Student's $t$ test. **e** Fasting glucose levels in *Gpr151* WT and KO DIO male mice measured in whole blood ($N = 10$, WT; $N = 9$, KO). Data are presented as mean values ±SEM. Two-tailed $t$ Student test. **f** Fasting insulin levels measured in blood plasma of DIO male mice ($N = 9$, WT; $N = 8$, KO). Data are presented as mean values ±SEM. Two-tailed Student's $t$ tests. **g–n** Metabolic phenotyping of DIO male mice using CLAMS, conducted at 23 °C. **g** Representative time course of oxygen consumption ($N = 5$, WT; $N = 5$, KO). **h** Average oxygen consumption ($N = 10$, WT; $N = 10$, KO). Data are presented as mean values ±SEM. Two-tailed Student's $t$ tests. **i** Representative time course of $CO_2$ production ($N = 5$, WT; $N = 5$, KO). **j** Average $CO_2$ production ($N = 10$, WT; $N = 10$, KO). Data are presented as mean values ±SEM. Two-tailed Student's $t$ tests. **k** Average respiratory exchange ratio ($N = 10$, WT; $N = 10$, KO). Data are presented as mean values ±SEM. Two-tailed Student's $t$ tests. **l** Average locomotion measured as average number of beam breaks per hour ($N = 9$, WT; $N = 8$, KO). Data are presented as mean values ±SEM. Two-tailed Student's $t$ tests. **m** Representative time course of energy expenditure ($N = 5$, WT; $N = 5$, KO). **n** Average energy expenditure ($N = 10$, WT; $N = 10$, KO). Data are presented as mean values ±SEM. Two-tailed Student's $t$ tests. *$p < 0.05$. Source data are provided as a Source Data file.

adipose tissue from the *Gpr151* KO mice compared to WT littermates in DIO, indicating liver-specific effects of *Gpr151* loss on energy metabolism (Fig. 3f). Further, the expression of genes encoding hepatic gluconeogenesis enzymes which are directly regulated by PPARGC1A, *Pck1* and *G6pc*[27], was also significantly decreased in *Gpr151* KO DIO livers (Fig. 3g). Further, western blotting confirmed the modest but significant downregulation of PEPCK in *Gpr151* KO livers at the protein level (Fig. 3h, i). Altogether, these data revealed that *Gpr151* loss leads to transcriptional downregulation of genes involved in glycolysis and gluconeogenesis in the liver that is consistent with PPARGC1A downregulation. However, more experiments will be needed to determine whether downregulation of PPARGC1A mediates the metabolic effects of *Gpr151* KO.

To functionally test the direct effect of *Gpr151* loss on hepatic gluconeogenesis, we assessed glucose and lactate production in DIO *Gpr151* WT and KO mice following an injection of pyruvate. In line with the insights from the RNA-Seq in the liver, pyruvate tolerance testing resulted in lower blood glucose levels in KO compared to WT mice, consistent with a lower gluconeogenesis action in the liver of *Gpr151* KO mice (Fig. 4a). Conversely, pyruvate administration resulted in nominal but not statistically significant elevation of blood lactate in *Gpr151* KO mice compared to WT littermates (Fig. 4b), further supporting a decrease in hepatic gluconeogenesis in *Gpr151* KO mice. Plasma levels of glucagon, the pancreatic hormone which stimulates hepatic gluconeogenesis[28], were not affected in *Gpr151* KO mice (Fig. 4c), supporting that the effects of *Gpr151* loss on glucose metabolism are selective to the liver. In addition, there were no changes in the expression of glucagon receptor in the liver between *Gpr151* WT and KO mice (Fig. 4d). To determine whether the impairment of hepatic gluconeogenesis by *Gpr151* loss is cell-autonomous, we assessed glucose secretion by glucagon-stimulated primary hepatocytes isolated from *Gpr151* WT and KO mice. *Gpr151* KO hepatocytes showed impaired glucose production compared to WT hepatocytes and did not increase glucose production in response to glucagon (Fig. 4e), demonstrating that *Gpr151* KO hepatocytes have a cell-autonomous impairment in both basal and glucagon-induced hepatic gluconeogenesis. In addition, gross liver morphology and the levels of liver triglycerides, plasma triglycerides, total cholesterol, HDL-cholesterol, and LDL-cholesterol were comparable in *Gpr151* KO mice and WT controls (Extended Data Fig. 6), indicating that in contrast to the differences in hepatic gluconeogenesis in *Gpr151* KO mice, there were no changes in liver lipid metabolism. In conclusion, we observe that loss of *Gpr151* directly impairs glucose production in primary hepatocytes ex vivo and in mice in a glucagon-independent way.

## cAMP-dependent gene expression is decreased in *Gpr151* KO livers

To identify the molecular mechanism explaining the impairment in hepatic gluconeogenesis in *Gpr151* KO hepatocytes, we next focused on the regulation of hepatic gluconeogenesis gene expression in the liver. CREB regulates transcription of hepatic gluconeogenesis genes and is activated by cAMP through protein kinase A (PKA)[26]. In glucagon-injected mice, the levels of CREB phosphorylation were nominally but not significantly lower in *Gpr151* KO mice compared to WT controls (Fig. 5a and Extended Data Fig. 7a). Additionally, there were no differences in insulin-dependent signaling between *Gpr151* WT and KO livers (Extended Data Fig. 7b–d), indicating specificity of the effect of *Gpr151* loss on glucagon signaling. To determine if there is orthogonal evidence for cAMP-dependent CREB activity dysregulation in the livers of *Gpr151* KO mice, we conducted a custom gene set enrichment analysis on the liver RNA-Seq data, using a set of genes which were previously identified as regulated by cAMP signaling in mouse hepatocytes[29] (Source Data). The analysis revealed a significant (p-val = $9.5 \times 10^{-8}$) enrichment of cAMP-regulated genes in WT livers, supporting an inhibition of cAMP-dependent transcription in *Gpr151* KO livers (Fig. 5b). CREB-regulated genes, such as *Dusp1*[30], *Ppp1r3c*[31], and *Btg1*[32] were significantly downregulated in *Gpr151* KO livers compared to WT (Fig. 5c). Therefore, GPR151 stimulates cAMP-dependent gene expression in the liver, including genes within hepatic gluconeogenesis pathway. In summary, *Gpr151* loss impairs hepatic gluconeogenesis regardless of the presence of glucagon, which likely explains its effects on whole-body glucose metabolism in diet-induced obesity (Fig. 5d).

## Liver overexpression increases glucose production in *Gpr151* KO

*Gpr151* loss leads to a decrease in basal and glucagon-dependent hepatic gluconeogenesis (Fig. 4e). To determine whether improved glucose metabolism in *Gpr151* KO mice is attributable to GPR151 function in the liver, recombinant adenovirus-associated virus serotype 8 (AAV8) was used to overexpress either *Gpr151* or green fluorescent protein (GFP) in the livers of DIO *Gpr151* KO mice (Fig. 6a). As expected, viral transduction was restricted to the liver, and elevated *Gpr151* transcript levels about 100-fold compared to livers from wild-type mice (Fig. 6b). Liver-specific overexpression of *Gpr151* did not lead to the expected decrease in glucose tolerance in *Gpr151* KO mice, although both groups of mice showed poor glucose tolerance (Extended Data Fig. 8).

Remarkably, pyruvate tolerance testing revealed a significant increase in the amount of glucose present in the blood following an injection of pyruvate in the mice with liver-specific *Gpr151* over-expression compared to *GFP*-overexpressing controls (Fig. 6c). This strongly supports a cell-autonomous role of liver GPR151 in hepatic glucose production important to regulate physiological glucose metabolism. Furthermore, several hepatic gluconeogenesis genes were upregulated following *Gpr151* over-expression, consistent with the increased blood glucose levels following pyruvate injection (Fig. 6d). In addition, the expression of the rate-limiting hepatic gluconeogenesis enzyme PEPCK in the liver increased significantly in the

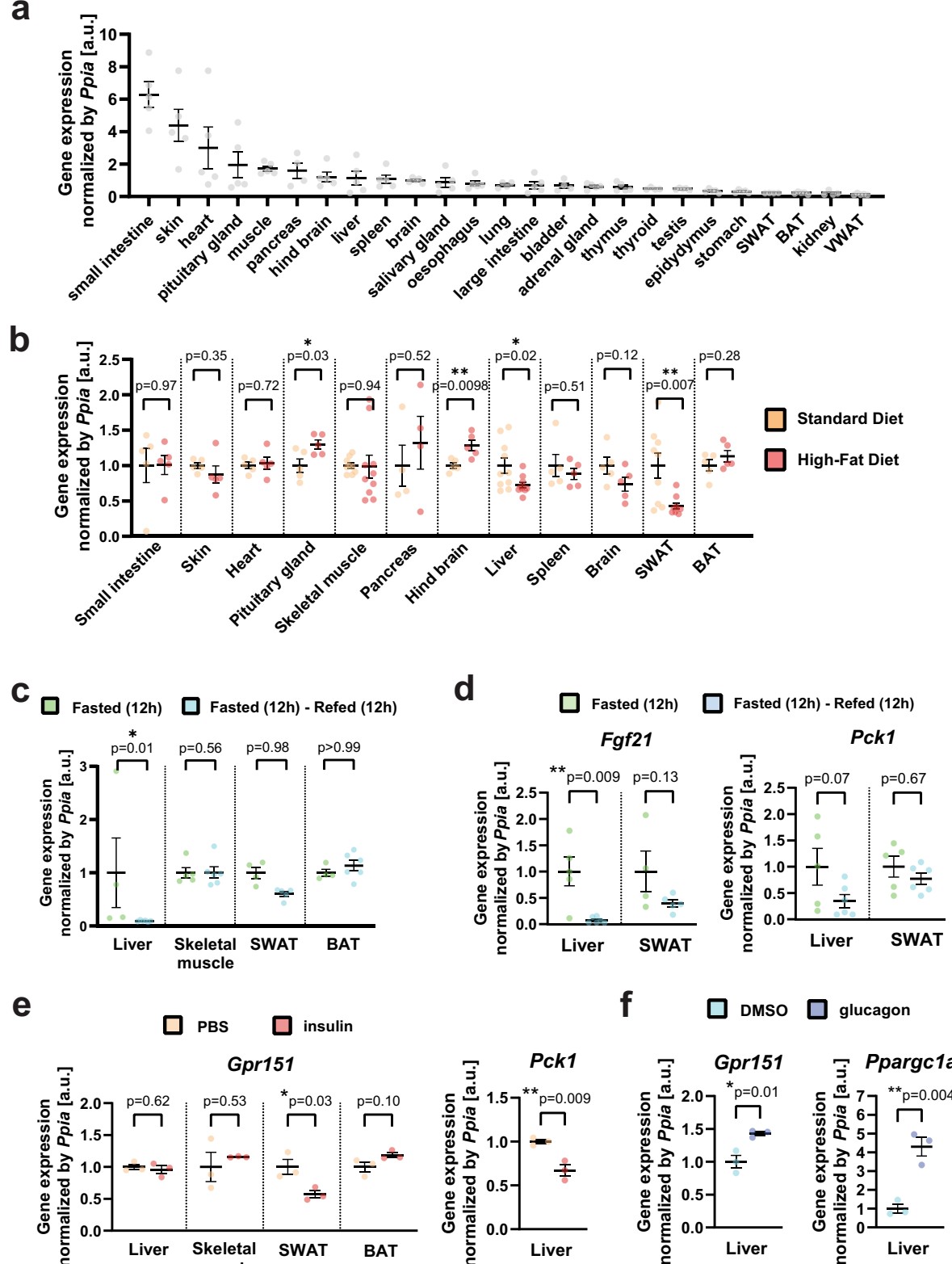

*Gpr151* KO mice injected with GPR151-expressing AAV8 compared to controls (Fig. 6e, f).

In summary, re-expression of *Gpr151* in the livers of whole-body *Gpr151* KO mice resulted in a reversal of some phenotypes associated with the defect in hepatic gluconeogenesis, a pathway that can be successfully targeted in the treatment of T2D[13]. Therefore, due to its function in the liver that we have uncovered, GPR151 appears to be a promising target for pharmacological regulation of blood glucose levels, although its role in relevant models of T2D remains to be studied.

**Fig. 2 | *Gpr151* expression is upregulated by fasting and glucagon in the liver.**
**a** RT-qPCR quantification of *Gpr151* expression in the tissues of eight-week-old male c57BL/6 J mice (*N* = 5). **b** RT-qPCR quantification of *Gpr151* expression in the 10 top-expressing tissues and the metabolically relevant tissues in HFD-fed mice compared to the same tissues from SD-fed mice. All mice were 16-week-old male c57BL/6 J mice (*N* = 4, SD; *N* = 4, HFD). Two-tailed Student's *t* tests. **c** RT-qPCR quantification of *Gpr151* expression in the liver, SWAT, BAT and skeletal muscle of mice which were fasted for 12 h and then refed for 12 h, normalized by *Gpr151* expression in the same tissues of mice fasted for 12 h. All mice were eight-week-old male c57BL/6J. Ordinary one-way ANOVA (*N* = 5, fasted; *N* = 5-6, refed). **d** Quantification of *Fgf21* expression in the liver and SWAT of fasted-refed and fasted mice. Ordinary one-way

ANOVA (*N* = 5, fasted; *N* = 5–6, refed). **e** *Gpr151* expression in the liver, skeletal muscle, SWAT and BAT in insulin-injected mice compared to the same tissues of PBS-injected mice, quantified by RT-qPCR. Quantification of *Pck1* expression in the liver shown as a positive control. Eight-week-old male c57BL/6J mice (*N* = 3, PBS-injected; *N* = 3, insulin-injected). Two-tailed Student's *t* tests. **f** *Gpr151* expression in the liver of glucagon-injected mice compared to the livers of control mice (vehicle-injected), quantified by RT-qPCR. Quantification of *Ppargc1a* expression in the liver shown as a positive control. Eight-week-old male c57BL/6J mice (*N* = 3, vehicle-injected; *N* = 3, glucagon-injected). Two-tailed Student's *t* tests. **a–f** All data are presented as mean values ±SEM. Source data are provided as a Source Data file.

### *GPR151* LOF variants are associated with favorable metabolic traits

Based on our observations in vitro and in vivo, we queried published GWAS studies for associations between the human *GPR151* p.Arg95Ter LOF variant (*rs114285050*, allele frequency 0.8% in European ancestry) and metabolic traits as summarized in Table 1. Previous studies in UK Biobank found that being a carrier of the *GPR151* LOF variant was associated with reduced BMI and lower risk of T2D[6,7]. Both associations were confirmed by larger independent GWAS studies[33,34]. In addition, there is indication of associations between the *GPR151* LOF variant with improved lipid profile (reduced triglycerides and increased HDL cholesterol)[35], as well as reduced waist-hip ratio adjusted by BMI (WHRadjBMI)[36] (all *p*-value < 0.05). Furthermore, there is indication of directionality that *GPR151* LOF is associated with improved glycemic traits (nominally reduced fasting glucose and reduced fasting insulin)[37]. These data strongly suggest that humans with a LOF *GPR151* variant have improved metabolic health.

### p.Arg95Ter *GPR151* variant results in the loss of the protein

Finally, to determine whether the p.Arg95Ter *GPR151* variant that is associated with lower risk of T2D leads to a functional loss of the GPR151 receptor, we conducted in vitro overexpression studies. While wild-type GPR151 receptor was consistently detected in the cellular membrane, the GPR151^Arg95Ter variant was undetectable, even though cells were transfected with comparable efficiency (Fig. 7, Extended Data Fig. 9). We conclude that while the wild-type GPR151 is localized in the cell membrane, the p.Arg95Ter variant likely results in the production of a truncated GPR151 protein that is degraded.

## Discussion

As the prevalence of IR and T2D keeps increasing worldwide, there is a need for the discovery of novel molecular targets and development of new therapies to target T2D, especially in those with insulin resistance. Here, we build on human genetics data to dissect the role of *Gpr151* in glucose metabolism and understand the mechanism behind the association between predicted LOF variants in *GPR151* and lower risk of T2D.

We report that *Gpr151* knockout in mice leads to cell-autonomous lowering of hepatic glucose production. This pathway is also targeted by the first-line antidiabetic drug metformin[38]. Although the molecular mechanism of metformin action in hepatocytes is not fully understood, it is known to affect mitochondrial respiration, leading to alterations in ATP:AMP ratios and inducing effects on AMPK signaling[39]. More recently AMPK-independent effects of metformin have also been identified[40]. Nevertheless, these pathways seem to be distinct from the GPR151 function. The mechanism of hepatic gluconeogenesis regulation by GPR151 should be explored further to understand whether it is relevant for the control of blood glucose levels in T2D. GPCRs are highly druggable molecular targets[41]. Therefore, GPR151 appears to be an exciting target candidate for the manipulation of hepatic gluconeogenesis.

In humans, the *GPR151* p.Arg95Ter LOF variant is associated with reduced BMI and WHRadjBMI, lower risk of T2D, improved lipid profile and a directionally consistent reduction of fasting glucose and insulin. However, we are limited to searching for associations in available published GWAS studies that may not be sufficiently powered to detect genotype effects of *GPR151* p.Arg95Ter due to its low allele frequency (minor allele frequency <1% in European populations), and therefore the reported effect sizes need careful interpretation. To quantify the clinical impact of the *GPR151* p.Arg95Ter LOF variant on metabolic traits, future recall-by-genotype studies of carriers of this variant should be performed. Such a trial might include detailed oral glucose tolerance tests as a measure of glucose homeostasis and T2D prevalence as primary outcomes. If designed properly, studies of human *GPR151* p.Arg95Ter LOF variant carriers may also provide information on molecular mechanisms of the role of GPR151 in its protective effect on metabolic traits.

Importantly, the role of *Gpr151* in the regulation of hepatic gluconeogenesis that we discovered is independent from the previously described role of *Gpr151* in the habenula, a brain structure that processes reward-related signals and affects appetite. Amongst peripheral metabolic tissues, *Gpr151* expression is regulated by feeding not only in the liver, but also in the white adipose tissue. Of note, the patterns of *Gpr151* expression in the liver and adipose tissue are distinct, indicating different mechanisms. In addition, liver-specific *Gpr151* overexpression reversed the hepatic gluconeogenesis phenotype in *Gpr151* KO mice but not the whole-body glucose tolerance. Therefore, while hepatic GPR151 regulates hepatic gluconeogenesis, this receptor likely has additional functions in other tissues that contribute to its effects on whole-body glucose metabolism. Whether GPR151 function in the fat contributes to whole-body glucose metabolism remains to be studied. In addition, *Gpr151* expression changes in adipose tissue in response to insulin could be secondary to blood glucose changes. Glucose clamp experiments would be required to discern between the effects of direct insulin action and changes in blood glucose levels.

In mouse hepatocytes, GPR151 is regulated at the level of gene expression. In particular, the expression of hepatic *Gpr151* is upregulated by fasting and the activity of the cAMP-inducing glucagon receptor and is at least partially mediated by CREB. Consequently, *Gpr151* loss would be predicted to primarily affect glucagon-induced hepatic gluconeogenesis. However, we instead observed that *Gpr151* KO hepatocytes show a general decrease in hepatic gluconeogenesis, independent of glucagon stimulation. This contradiction may be due to a chronic downregulation of hepatic gluconeogenesis in *Gpr151* KO hepatocytes and is supported by the decrease in hepatic gluconeogenesis when hepatic Gi signaling is inhibited in general[19]. Molecular mechanism underlying this paradoxical gluconeogenesis impairment by the loss of cAMP-inhibiting GPCRs remains to be understood.

Taken together, we show here that GPR151 regulates hepatic gluconeogenesis. In the future, development of molecular tools, in particular of an inverse agonist to GPR151, would allow for assessment of the feasibility of targeting this receptor to control blood glucose levels in IR and T2D models.

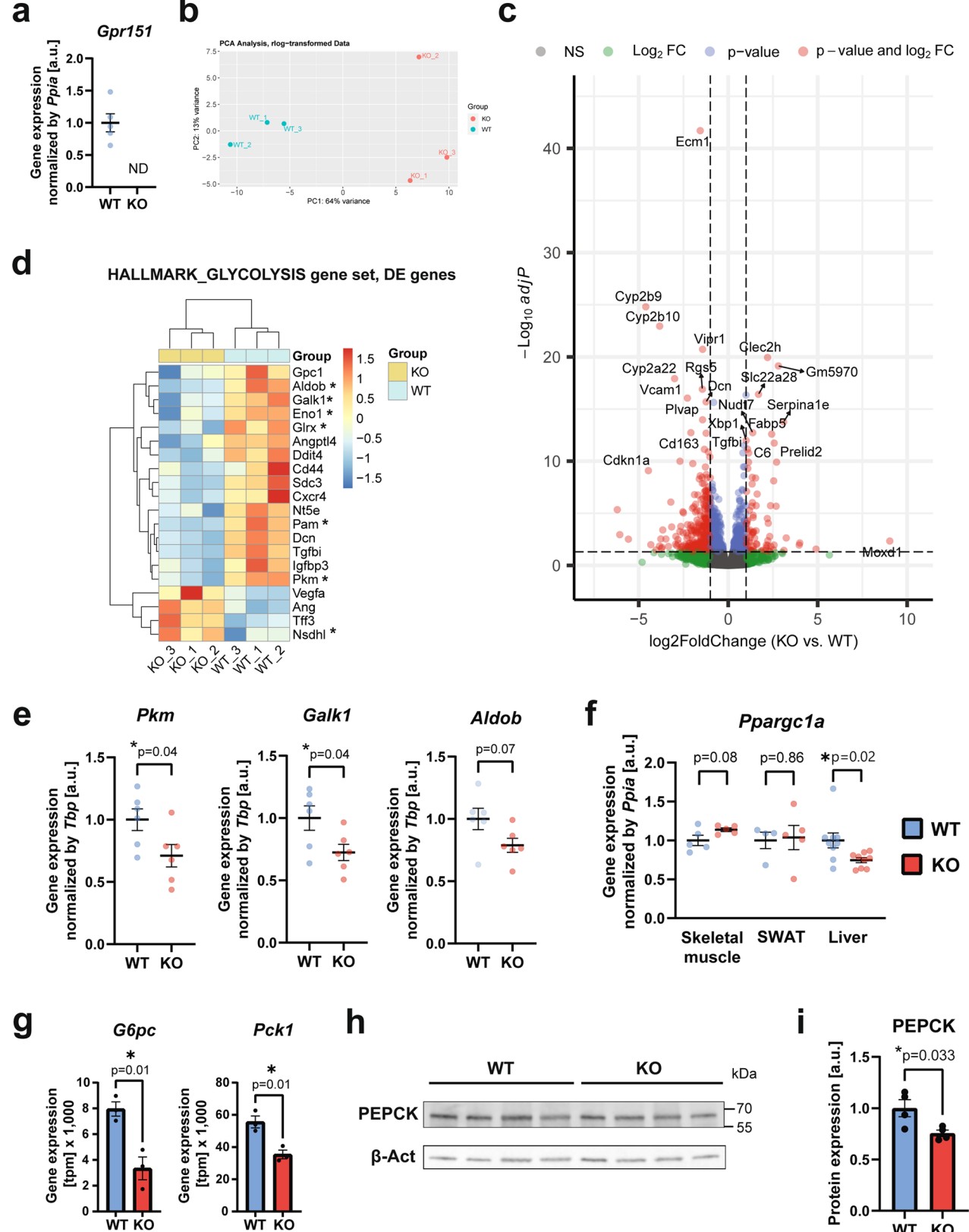

## Methods

### Animals

*Gpr151* KO mice on the C57BL/6 J genetic background, described previously[10], were generously provided by Dr. Ibañez-Tallon from the Rockefeller University. The mice were crossed with C57BL/6 J mice (#000664, Jax) to obtain heterozygous animals which were further in-crossed. For experiments conducted exclusively on wild-type mice, male C57BL/6 J mice (#000664, Jax) were purchased. All animal studies were approved by the Administrative Panel on Laboratory Animal Care at Stanford University, and were performed according to the guidelines of the American Association for the Accreditation of Laboratory Animal Care.

**Fig. 3 | GPR151 upregulates the expression of hepatic gluconeogenesis genes.**
**a** RT-qPCR quantification of *Gpr151* gene expression in the livers of *Gpr151* WT and KO mice (*N* = 5, WT; *N* = 5, KO). Data are presented as mean values ±SEM. **b** Principal component analysis showing the clustering of samples analyzed by RNA-Seq. **c** Volcano plot showing the statistical significance (−log10(p$_{adj}$)) versus log2 of fold change of gene expression between *Gpr151* KO and WT livers. **d** Heat map of transcriptional regulation patterns between DE genes (*p*-val$_{adj}$ < 0.05) from the glycolysis/gluconeogenesis pathway within the HALLMARK gene sets. Genes which encode enzymes are marked with an asterisk. **e** RT-qPCR quantification of selected genes from the glycolysis/gluconeogenesis pathway in 16-week-old DIO female mice (*N* = 6, WT; *N* = 6, KO). Data are presented as mean values ±SEM. Two-tailed Student's *t* tests. **f** RT-qPCR quantification of *Ppargc1a* expression in the liver,

skeletal muscle and adipose tissue of *Gpr151* WT and KO DIO male mice (*N* = 4, WT; *N* = 4, KO). Data are presented as mean values ±SEM. Two-tailed Student's *t* tests. **g** Quantification of the expression of PPARGC1A target genes *G6pc* and *Pck1* in *Gpr151* WT and KO livers by RNA-Seq. *N* = 3 mice/group. Data are presented as mean values ±SEM. Two-tailed Student's *t* tests (*N* = 3, WT; *N* = 3, KO). **h** Representative Western blotting of PEPCK in the livers of 16-week-old DIO *Gpr151* WT and KO male mice. Samples were run on the same blot. Results of one experiment representative for two independent experiments. **i** Quantification of PEPCK to β-actin in the livers of 16-week-old DIO *Gpr151* WT and KO mice (*N* = 4, WT; *N* = 4, KO). Average and S.E.M. are indicated. Two-tailed Student's *t* test. **p* < 0.05. Source data are provided as a Source Data file.

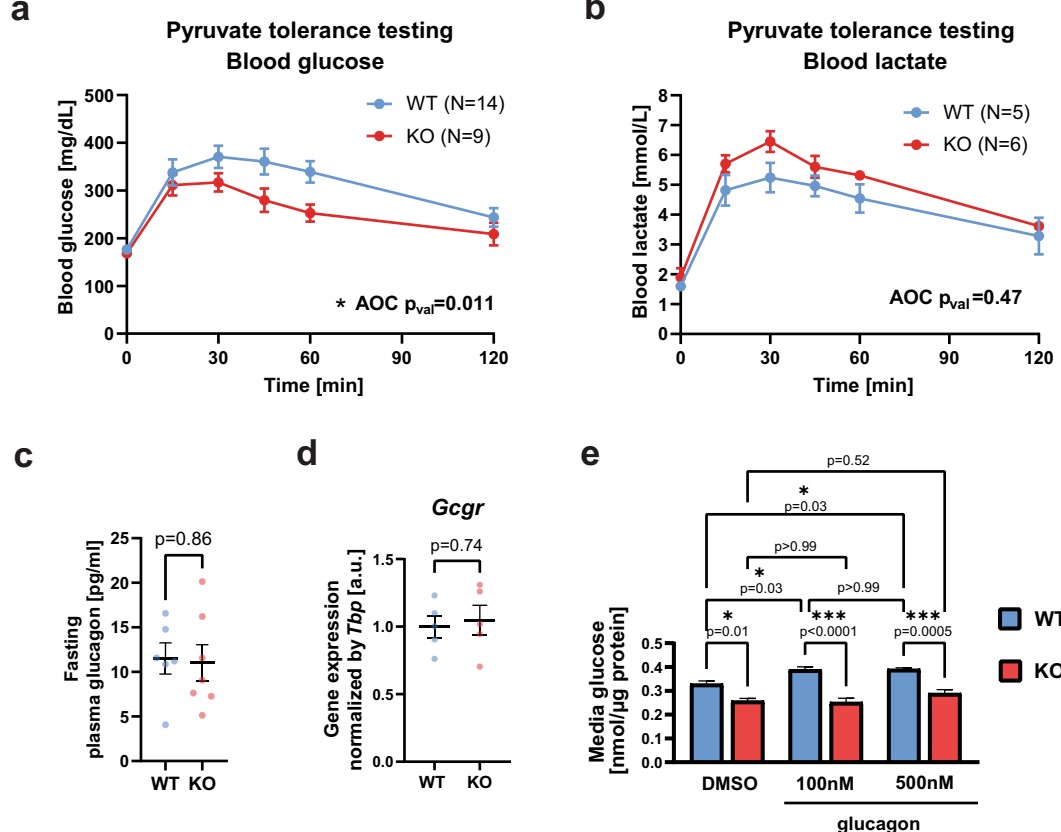

**Fig. 4 | *Gpr151* regulates hepatic gluconeogenesis in a cell-autonomous manner.**
**a** Blood glucose levels measured during pyruvate tolerance testing (PTT) in male DIO WT and KO mice. AOC compared using two-tailed Student's *t* test (*N* = 14, WT; *N* = 9, KO). **b** Blood lactate levels during PTT in male DIO WT and KO mice. AOC compared using two-tailed Student's *t* test (*N* = 5, WT; *N* = 6, KO). **c** Quantification of glucagon levels in blood plasma of fasted 16-week-old DIO mice by ELISA. Two-tailed Student's *t* test (*N* = 6, WT; *N* = 7, KO). **d** RT-qPCR quantification of glucagon

receptor (*Gcgr*) gene expression in the livers from *Gpr151* WT and KO DIO male mice. Two-tailed Student's *t* test (*N* = 5, WT; *N* = 5, KO). **e** Quantification of media glucose produced by *Gpr151* WT and KO primary hepatocytes ex vivo, normalized by protein concentration. Results of one experiment representative for three independent experiments. Ordinary one-way ANOVA with Sidak's multiple comparisons test (p$_{val}$ < 0.0001). *n* = 3 technical replicates. **a**–**e** All data are presented as mean values ±SEM. Source data are provided as a Source Data file.

Unless indicated otherwise, mice were housed with *ad libitum* access to chow and water in an air-conditioned room with a standard 12-hour light/dark cycle with relative humidity of 30–70% and temperature of 20–26 °C. For studies on diet-induced obesity, mice were fed a standard diet (18% protein / 6% fat, #2918, Envigo Teklad) for the first 4 weeks, followed by either high-fat diet (HFD, 60 kcal% Fat, D12492, Research Diets) or control sucrose-matched diet (SD, 10 kcal% Fat, D12450J, Research Diets) for the remainder of the experiment. Body weight and chow intake were monitored weekly.

To quantify the changes in gene expression in response to insulin in vivo, eight-week-old c57BL/6 J male mice were fasted for 4 h, injected i.v. with either PBS or insulin (Humulin, Eli Lilly) at 1 U/kg body

weight and sacrificed after 4 h. To assess protein phosphorylation in response to insulin, 16-week-old male *Gpr151* WT and KO mice fed HFD for 12 weeks were injected i.v. with 1 U/kg body weight (Humulin, Eli Lilly) diluted in PBS or vehicle control (PBS) and sacrificed after 10 min. To assess gene expression changes in response to glucagon, eight-week-old male c57BL/6J mice were injected i.p. with 2 mg/kg body weight glucagon (Sigma-Aldrich, #G2044) in PBS or vehicle control and sacrificed after 60 min. To assess protein phosphorylation in response to glucagon, eight-week-old male *Gpr151* WT and KO mice fed HFD for 4 weeks were injected i.v. with either 2 mg/kg body weight glucagon (Sigma-Aldrich, #G2044) diluted in PBS or vehicle control (DMSO in PBS) and sacrificed after 10 min.

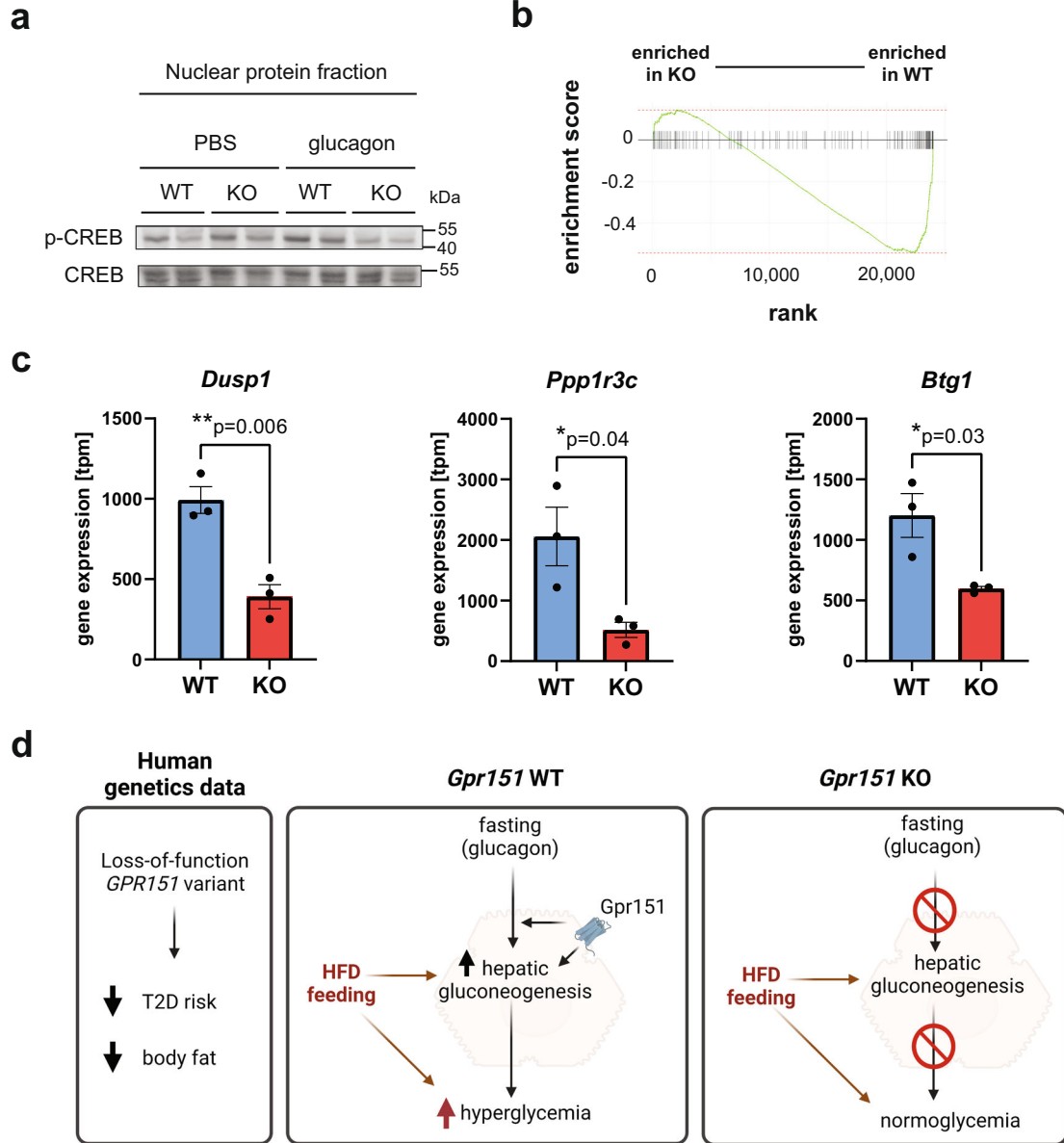

**Fig. 5 | *Gpr151* loss results in a downregulation of cAMP-dependent gene expression. a** Western blotting of phosphorylated CREB and total CREB in the nuclear protein fraction isolated from the livers of PBS- and glucagon-injected *Gpr151* WT and KO eight-week-old male mice. Samples were run on the same blot. Results of one experiment representative of two independent experiments. **b** The results of custom gene set enrichment analysis of liver expression of 127 cAMP-responsive genes in the liver of *Gpr151* WT and KO mice. **c** Quantification of the expression of CREB-regulated genes *Dusp1, Igfbp1, Btg1* in *Gpr151* WT and KO livers by RNA-Seq. $N = 3$ mice/group. Data are presented as mean values ±SEM. Two-tailed Student's *t* tests ($N = 3$, WT; $N = 3$, KO). **d** A model of the role of GPR151 in metabolic health. Schematic created using Biorender. Source data are provided as a Source Data file.

### Indirect calorimetry

For measurements of metabolic rate and food intake, approximately 16-week-old male mice fed HFD for 12 weeks were placed within the CLAMS (Columbus Instruments) indirect calorimeter. Prior to the experiment, body composition of conscious mice was assessed with an EchoMRI 3-in-1 (Echo Medical Systems). Mice were acclimated to CLAMS for 13 h followed by 48 h measurement of $VO_2$, $VCO_2$, RER, locomotor and ambulatory activity, food intake at $23 \pm 0.1\,°C$ while on HFD. Energy expenditure was calculated as previously described[42]. Calories consumed were calculated by multiplying hourly food intake by the 5.21 kcal/g caloric value of the 60% HFD. Energy balance was calculated by subtracting hourly food intake from hourly energy expenditure. Data from two separate experimental runs were combined. Analyses using ANOVA and ANCOVA were performed using CalR without the remove outliers feature[43].

### Glucose, insulin, and pyruvate tolerance testing

GTTs, ITTs and PTTs were performed on 16- to 17-week-old mice. GTTs were performed after an overnight fast (14–16 h). 1.5 g glucose / kg body weight was injected i. p. ITTs were performed after 5-6 h fasting. Mice were injected i.p. with insulin (Humulin, Eli Lilly) with various concentrations depending on the group (males receiving standard diet – 0.75 U/kg body weight, females receiving standard diet – 0.5 U/kg body weight, mice receiving high-fat diet – 1 U/kg body weight). PTTs were performed after an overnight fast (16–18 h). 1 g sodium pyruvate / kg body weight was injected i.p. Blood glucose levels were measured from the tail vain using a glucometer (TRUEbalance, Nipro Diagnostics), and blood lactate levels were measured using Lactate Plus Meter (Nova Biomedical). For statistical comparisons, the area of the curve (AOC)[44] was calculated after subtracting baseline level of the metabolite measured, followed by statistical testing using Student's *t*

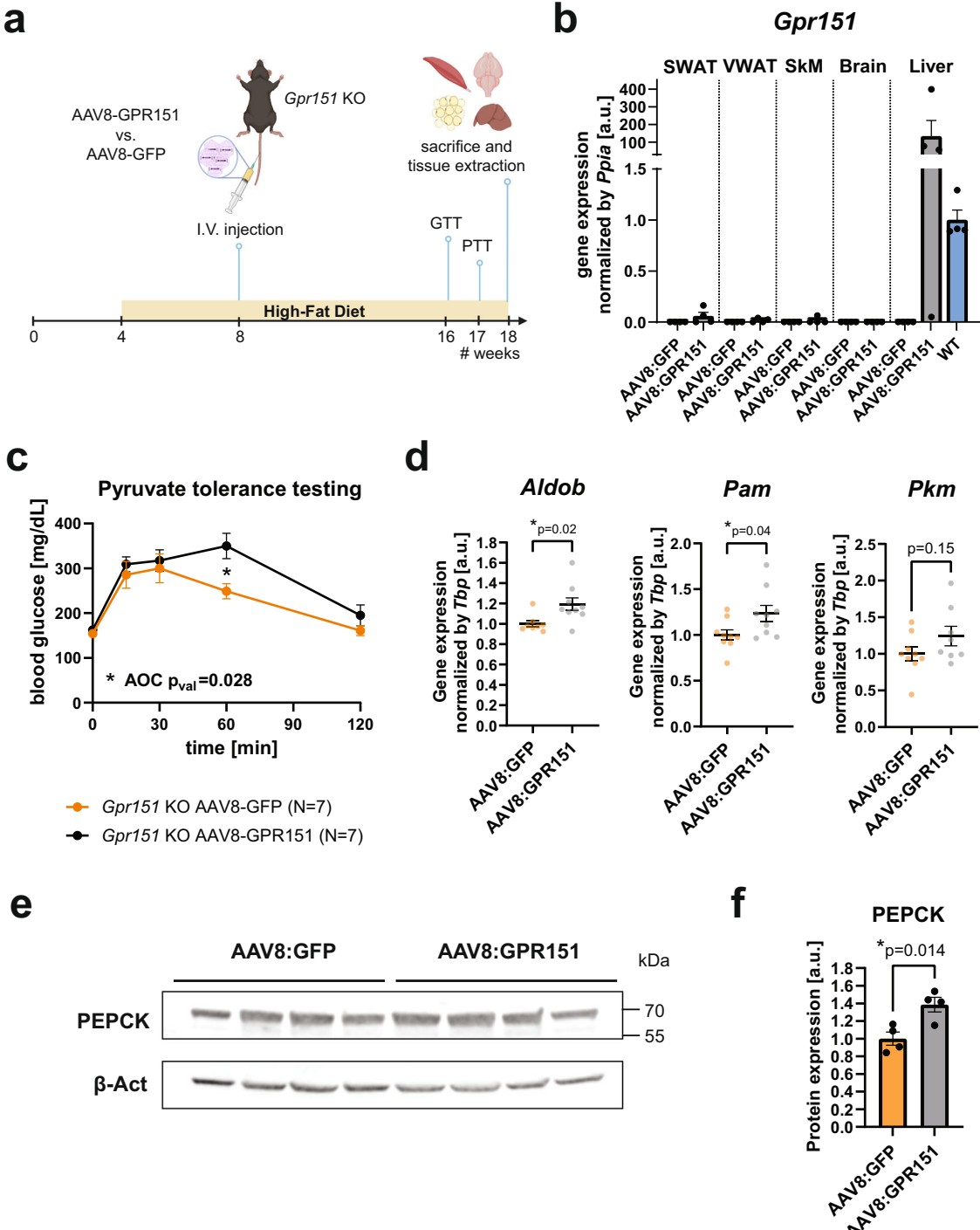

**Fig. 6 | Overexpression of *Gpr151* in the liver of DIO *Gpr151* KO mice rescues the downregulation of hepatic gluconeogenesis gene expression. a** Schematic of the experiments involving liver overexpression of *Gpr151* using AAV8, created using Biorender. **b** Quantification of *Gpr151* overexpression by RT-qPCR in subcutaneous (SWAT) and visceral (VWAT) adipose tissue, skeletal muscle (SkM), brain and liver of *Gpr151* KO mice injected either with AAV8-GFP or AAV8-GPR151. Data normalized to *Gpr151* expression in the livers of age-, sex- and diet-matched wild-type mice (*N* = 4, *Gpr151* KO AAV8:GFP; *N* = 4, *Gpr151* KO AAV8:GPR151). Data are represented as mean values + SEM. **c** Blood glucose levels in *Gpr151* KO DIO mice injected with AAV8-GPR151 or AAV8-GFP, subjected to pyruvate tolerance testing. Data are represented as mean values ±SEM. AOC compared using two-tailed Student's *t* test. Two-tailed Student's *t* test with Bonferroni correction used to test differences at every time point (\**t* = 60 $q_{val}$ = 0.049; *N* = 7 *Gpr151* KO AAV8:GFP; *N* = 7, *Gpr151* KO

AAV8:GPR151). **d** RT-qPCR quantification of the expression of hepatic gluconeogenesis genes, identified as downregulated in *Gpr151* KO mice compared to WT, in the livers of *Gpr151* KO mice with liver-specific GPR151 and GFP overexpression by AAV8 (*N* = 8–9, *Gpr151* KO AAV8:GFP; *N* = 8–9, *Gpr151* KO AAV8:GPR151). Data are represented as mean values ±SEM. Two-tailed Student's *t* test. **e** Western blotting of PEPCK in the livers of *Gpr151* KO mice with liver-specific GPR151 and GFP overexpression by AAV8. Samples were run on the same blot. Results of one experiment representative for two independent experiments. **f** Quantification of PEPCK to β-actin in the livers of *Gpr151* KO mice with liver-specific GPR151 and GFP overexpression by AAV8 (*N* = 4, *Gpr151* KO AAV8:GFP; *N* = 4, *Gpr151* KO AAV8:GPR151). Average and S.E.M. are indicated. Two-tailed Student's *t* test; \**p* < 0.05. Source data are provided as a Source Data file.

**Table 1 | Results from query of GWAS studies with publicly available summary statistics**

| Trait | N in thousands (K) | Beta | Directionality | P-value | Ancestry | Reference |
|---|---|---|---|---|---|---|
| Genome-wide significant associations (P < 5×10⁻⁸) | | | | | | |
| BMI | 700 K | −0.064 | – | $3.7 \times 10^{-9}$ | European | Yengo et al.[33] |
| Suggestive associations (P < 0.05) | | | | | | |
| Type 2 diabetes | 898 K | −0.093 | – | 0.021 | European | Mahajan et al.[34] |
| Triglycerides | 608 K | −0.024 | – | 0.044 | Multi-ancestry | Klarin et al.[35] |
| HDL cholesterol | 608 K | 0.0246 | + | 0.033 | Multi-ancestry | Klarin et al.[35] |
| WHRadjBMI | 694 K | −0.031 | – | 0.004 | European | Puilt et al.[36] |
| Directionally consistent associations | | | | | | |
| Fasting glucose[a] | 80 K | −0.0067 | – | 0.84 | European | Chen et al.[37] |
| Fasting insulin[a] | 67 K | −0.0121 | – | 0.47 | European | Chen et al.[37] |

Directionality indicates the effect of the *GPR151* p.Arg95Ter loss-of-function variant (*rs114285050* A-allele is the effect allele, while the G-allele is the other allele) on the respective trait analyzed.
*GPR151 rs114285050* chromosomal position is Chr5:146515831 (GRCh38).
*N* number of participants of respective GWAS study.
[a]GWAS analyses were adjusted for BMI.

test. If the difference in AOC was statistically significant, Student's *t* test with Bonferroni correction was used for the follow-up comparison of metabolite levels at different time points.

## Systemic metabolic parameters

Blood was obtained by collection from vena cava from euthanized animals which had been fasted for 5–6 h. Heparin (Sigma Alrich, #H3393) solution in PBS was used to wash syringe. Collected blood was immediately centrifuged at room temperature to obtain plasma. ELISA was used to measure plasma glucagon (Mouse glucagon ELISA kit, Crystal Chem, #81518) and insulin (Ultra Sensitive Mouse Insulin ELISA kit, Crystal Chem, #90080). Plasma triglycerides, HDL and LDL were measured by the Diagnostic Laboratory at the Department of Comparative Medicine at Stanford University.

## *Gpr151* overexpression in vivo

Adeno-associated viruses serotype 8 (AAV8) encoding *GFP* (AAV8-GFP) and *Gpr151* (AAV8-GPR151) were purchased from Vector Biolabs. HFD-fed *Gpr151* KO male mice were injected intravenously with $5 \times 10^{10}$ vg / mouse at 8 weeks of age. Following the injection, GTT was performed at 16 weeks of age and PTT was performed at 17 weeks of age. Mice were sacrificed at 18 weeks of age.

## Cell culture

AML12 mouse hepatocyte cell line was obtained from ATCC and cultured in Dulbecco's Modified Eagle Medium/Nutrient Mixture F-12 (Invitrogen, #11330057) with the addition of 10% Fetal Bovine Serum (BenchMark Fetal Bovine Serum, GeminiBio, #100–106), 1x Insulin-Transferrin-Selenium (ITS-G, Gibco, #41400045), 40 ng/ml dexamethasone (Sigma, #D4902), and 1x Pen/Strep (Thermo Fisher Scientific, #15140163). Cells were cultured in a humidified 5% $CO_2$ incubator. Serum starvation was carried out overnight in DMEM/F-12 with the addition of 1x Pen/Strep.

Primary hepatocytes were isolated from eight-week-old male mice and cultured as previously described[45]. For each sample, hepatocytes isolated from two mice of the same genotype were pooled. All experiments were performed within 36 h of hepatocyte isolation.

HEK293T cells were obtained from ATCC (CRL-3216) and cultured in DMEM media (Thermo Fisher Scientific) with the addition of 10% Fetal Bovine Serum and 1x Pen/Strep. For transfection, cells were seeded in 96-well glass bottom plates (Ibidi, #89626) and transfected with plasmids using Lipofectamine 3000 Transfection Reagent (Thermo Fisher Scientific) according to the manufacturer's protocol. 24 h after transfection, cells underwent fixation.

Cell were stimulated with 666-15 (Tocris, #5661), forskolin (Sigma Aldrich, #F6886), glucagon (Sigma Aldrich, #G2044), dexamethasone (Sigma Aldrich, #D4902).

Media glucose was quantified using Glucose Assay Kit (Abcam, ab65333) according to manufacturer's protocol. Glucose levels were normalized using Pierce BCA Protein Assay Kit (Thermo Fisher Scientific, #23209).

## Immunofluorescent staining and confocal imaging

Cells were fixed in 4 % paraformaldehyde/PBS for 15 min at room temperature, followed by three washes in PBS. Permeabilization was achieved by incubation in 0.2 % Triton X-100/PBS for 15 min on ice, followed by three PBS washes. Cells were incubated in blocking solution (3% Bovine Serum Albumin/PBS) overnight, followed by staining with anti-HA antibody conjugated to AlexaFluor647 (BioLegend, #682404, 1:1000) and Hoechst (1:2000) in blocking solution overnight. Finally, cells were washed three times in PBS and imaged using confocal microscopy. Images were collected on a Yokogawa spinning disc confocal on a Nikon Eclipse-Ti inverted microscope (Nikon) equipped with a PLAN APO 60×1.40 N.A. oil immersion objective. Images were acquired with a Photometrics Prime 95B sCMOS camera controlled with NiS-Elements software, using $1 \times 1$ binning, in single z-plane. Images were exported and fluorescence in the EGFP (490 nm) and Cy5 (645 nm) channels was quantified manually using ImageJ 1.53c software.

## Liver triglyceride measurement

Liver triglycerides were quantified using Triglyceride Assay Kit (Abcam, ab65336) according to manufacturer's protocol, and normalized by tissue sample weight.

## Histology

For hematoxylin and Eosin (H&E) staining tissues were harvested, weighed and fixed in 10% Neutral Buffered Formalin (Sigma Aldrich, #HT501320) for 72 h, following by dehydration in 70% ethanol. Dehydrated tissues underwent standard H&E staining at the Stanford Animal Histology facility, using the following steps: xylene (2 min × 3), 100% ethanol (2 min), 100% ethanol (1 min), 95% ethanol (1 min × 2), 80% ethanol (1 min), running tap water (1 min), Harris hematoxylin pH 2.5 (10 min), running tap water (1 min), 1% HCl/70% ethanol (20 s), running tap water (5 min), 0.5% ammonium hydroxide/deionized water, running tap water (3 min), 95% ethanol (1 min), eosin (95% ethanol solution pH 4.6, 2 min), 95% ethanol (30 s x 3), 100% ethanol (2 min x 3), xylene (2 min x 3), and sealed with Cytoseal XYL.

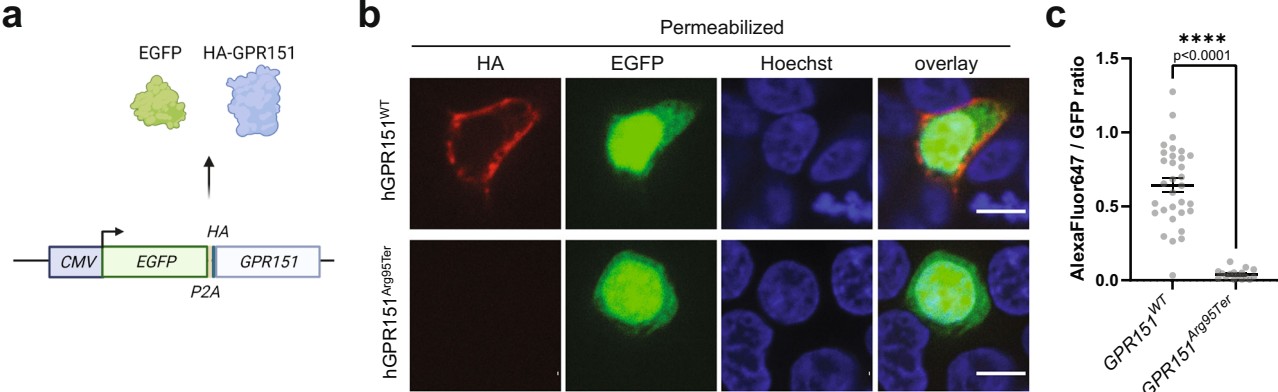

**Fig. 7 | Cellular localization of the wild-type and p.Arg95Ter GPR151.**
**a** Schematic of the overexpression plasmids used. EGFP and HA-tagged GPR151 are produced in equal ratio from a single transcript. Image created using Biorender. **b** Confocal images of EGFP-positive HEK293T cells that were transfected with each of the plasmids, fixed, permeabilized and fluorescently stained against the HA tag. Results of one experiment representative for three independent experiments.

Nuclei counterstained with Hoechst. Images from a single z-plane, 60x objective. Scale bar: 10 μm. **c** Quantification of the ratio between total HA staining (Alexa-Fluor647) and total GFP signal over the entire cell area in a single confocal z-plane ($n = 32$, GPR151$^{WT}$; $n = 17$, GPR151$^{Arg95Ter}$). Average and S.E.M. are indicated. Two-tailed Student's $t$ test; ****$p < 0.0001$. Data are presented as mean values ±SEM. Source data are provided as a Source Data file.

Histological images were collected using Zeiss Axioplan2 microscope at 20x magnification using Leica DC500 camera and NIS Elements software.

## RNA analyses

Total RNA was extracted from in vitro cell cultures using RNAeasy Mini kit (QIAGEN, #74106) or from tissue using Ambion TRIzol Reagent (Thermo Fisher Scientific, #15-596-018) according to the manufacturer's instructions. To remove any contaminating DNA, RNA was treated with Invitrogen TURBO DNA-free Kit (Invitrogen, #AM1907), according to the manufacturer's instructions. Next, RNA was converted to cDNA using High-Capacity cDNA Reverse Transcription Kit (Applied Biosystems, #4374966). Quantitative PCR was conducted using TaqMan Fast Advanced Master Mix (Thermo Fisher Scientific, #4444557) and performed on ViiA 7 Real-Time PCR System (Thermo Fisher Scientific). All data were normalized to the expression of the housekeeping genes cyclophilin A (*Ppia*) or TATA-box binding protein (*Tbp*). The TaqMan assays (Integrated DNA Technologies) were: *Adra2a* (Mm.PT.58.33590743.g), *Adra2b* (Mm.PT.58.6098909.g), *Adra2c* (Mm.PT.58.30363072.g), *Aldob* (Mm.PT.58.42159810), *Cnr1* (Mm.PT.58.30057922), *Fgf21* (Mm.PT.58.29365871.g), *G6pc* (Mm.PT.58.11964858), *Galk1* (Mm.PT.58.7143435), *Gcgr* (Mm.PT.58.16192096), *Gpr151* (Mm00808987_s1), *Pam* (Mm.PT.58.8164623), *Pck1* (Mm.PT.58.11992693), *Pkm* (Mm.PT.58.6642152), *Ppargc1a* (Mm.PT.58.28716430), *Ppia* (Mm02342430_g1), and *Tbp* (Mm.PT.39a.22214839). *LacZ* Taqman Gene Expression Assay was ordered from Thermo Fisher Scientific (Mr03987581_mr).

For RNA-Seq, RNA was extracted from snap-frozen liver tissue using TRIzol according to manufacturer's protocol. Library preparation and sequencing was conducted by Novogene using the mRNA-Seq pipeline, with over 20 million raw reads per sample and Q30 of over 96%. Raw paired-end FASTQ files were filtered to remove reads with adapter contamination, reads when uncertain nucleotides constitute more than 10% of either read ($N > 10\%$), and reads when low quality nucleotides (Base Quality less than 5) constitute more than 50% of the read. STAR 2.6.1d[46] was used to align clean reads to the mouse reference genome (Ensembl Mus Musculus GRCm38.p6 GCA_000001635) and to generate gene counts. STAR output for each sample was combined for differential expression testing using DESeq2_1.32.0 R package installed in R v4.1, using default settings. PCA was performed on rlog-transformed data using DESeq2[47]. Gene set enrichment analysis was performed using GSEA v4.1.0 on Hallmark gene sets[25] with

Permutation type set to Gene set. Additionally, gene set enrichment analysis was performed on a custom gene set of 127 cAMP-responsive genes[29] using fgsea v1.18.0[48]; specifically, the fgsea-Multilevel method was used with default settings. Both gene set enrichment analyses were performed using DESeq2 normalized counts.

## Western blotting

Tissue samples were snap-frozen in liquid nitrogen, ground using mortar and pestle and lysed in T-PER Tissue Protein Extraction Reagent (Thermo Fisher Scientific, #78510) with Halt Protease and Phosphatase Inhibitor Cocktail (Thermo Fisher Scientific, #78445). Nuclear and cytoplasmic protein fractions were isolated using Nuclear Extraction kit (Abcam, ab219177) according to manufacturer's protocols. Protein concentrations were determined using Pierce BCA Protein Assay Kit (Thermo Fisher Scientific). Samples were subjected to SDS-PAGE in polyacrylamide gels (4–20% Mini-Protean TGX, Bio-Rad, #4561094) and transferred onto 0.45 μm PVDF membrane (Thermo Fisher Scientific, #88518) using Mini-PROTEAN system (Bio-Rad). Proteins were detected using primary antibodies against CREB (clone 48H2, Cell Signaling, #9197, 1:1000), p-CREB (Millipore Sigma, #06-519, 1:1000), Akt (clone E7J2C, Cell Signaling, #58295, 1:1000), phospho-Akt (clone D9E, Cell Signaling, #4060, 1:1000), PEPCK (Abcam, ab70358, 1:1000), S6 (clone 54D2, Cell Signaling, #2317, 1:1000), and phospho-S6 (Cell Signaling, #2211, 1:1000). Equal protein loading was verified using antibody against β-actin (clone C-4, Santa Cruz, sc-47778, 1:15,000). Secondary antibodies were: anti-rabbit IRDye 680RD (goat anti-rabbit IgG, #925-68071, Li-cor, 1:10,000), anti-mouse IRDye 800CW (goat anti-mouse IgG, #925-32210, Li-cor, 1:10,000), anti-mouse HRP (Cell Signaling, #7076, 1:10,000), anti-rabbit HRP (Cell Signaling, #7074, 1:10,000). HRP was visualized using SuperSignal West Pico PLUS Chemiluminescent Substrate (Thermo Fisher Scientific, #34579) and Supersignal West Femto Maximum Sensitivity substrate (Thermo Fisher Scientific, #34096) according to manufacturer's protocol. Blots were imaged using Odyssey Fc imager (Li-cor). Protein expression was quantified using ImageJ 1.53c. Uncropped images of blots are provided in Extended Data Fig. 10.

## Molecular cloning

CMV:eGFP-p2A-HA-β2Ar-uTEV1Δ(220-242) vector[49], which was a gift from Alice Ting (Addgene plasmid # 135459; http://n2t.net/addgene:135459; RRID:Addgene_135459), was digested using AgeI and MluI (New England Biolabs), and the backbone was purified from agarose

gel using the QIAquick Gel Extraction Kit (QIAGEN). Gibson assembly was conducted using custom gBlock sequences (Integrated DNA Technologies), encoding either GPR151[WT] or GPR151[Arg95Ter], and the Gibson Assembly Master Mix (New England Biolabs), according to the manufacturer's protocol. Correct sequences of the resulting plasmids were verified using Sanger sequencing.

## Analysis of promoter sequences

Conserved predicted transcription factor-binding sites were identified using ConTra v3[50] by the analysis of 1,000 bp promoter sequence upstream of human *GPR151* (ENST00000311104) transcript. Initial search was conducted using position weight matrices (PWMs) for glucocorticoid receptor (GR TRANSFAC20113,V\$GR_Q6,M00192), FOXO1 (FOXO1 TRANSFAC20113,V\$FOXO1_Q5,M01216), and CREB (CREB TRANSFAC20113,V\$CREB_Q4_01,M00917) using core = 0.95, similarity matrix = 0.85 settings.

## Query of published summary statistics from GWAS analysis in humans

We queried summary statistics for GWAS of BMI and T2D to confirm previous reports. We queried selected GWAS of selected traits (WHRadjBMI, triglycerides, HDL cholesterol, fasting glucose and fasting insulin) to explore if our findings in vivo translate to human variant carriers. Summary statistics for GWAS of BMI, WHRadjBMI, fasting glucose and fasting insulin were downloaded from GWAS catalog with the following accession codes: GCST006900 (BMI)[33], GCST008996 (WHRadjBMI)[36], GCST90002232 [https://www.ebi.ac.uk/gwas/studies/GCST90002232](fasting glucose)[37], GCST90002238 (fasting insulin)[37]. The summary statistics from the GWAS of T2D[34] were downloaded from the DIAGRAM consortium website (http://diagram-consortium.org/), and the summary statistics from the GWAS of triglycerides and HDL cholesterol[35] were downloaded through dbGaP, with accession number phs001672.v1.p1. Association results for the *GPR151* p.Arg95-Ter LOF variant was retrieved based on rs-number (*rs114285050*) or position (Chr5:145895394 for GRCh37, Chr5:146515831 for GRCh38). All GWAS studies were performed assuming additive genetic effects. We aligned the effect allele of the *rs114285050* across studies (A-allele is the effect allele; G-allele is the other allele) to indicate directionality of effect for *GPR151* p.Arg95Ter loss-of-function variant carriers.

## Statistical analysis

Unless indicated, all data are expressed as mean ± standard error of the mean (SEM). Student's *t* test was used for single variables and one-way ANOVA with Bonferroni post hoc correction (or equivalent) was used for multiple comparisons using GraphPad Prism 7 software.

## Reporting summary

Further information on research design is available in the Nature Portfolio Reporting Summary linked to this article.

# Data availability

The raw transcriptome data generated in this study have been deposited with the NCBI's Gene Expression Omnibus under accession code #GSE196535. Overexpression plasmids generated in this study were deposited with the Addgene depository (IDs 190133 and 190134). Summary statistics for GWAS were downloaded from https://www.ebi.ac.uk/gwas/ with accession codes GCST006900, GCST008996, GCST90002232, GCST90002238, from http://diagram-consortium.org/ (data described as "T2D GWAS meta-analysis – Genetic Credible Sets"), and dbGaP with accession number phs001672.v1.p1. All data supporting the findings of this study are available within the article and Supplementary Information files. Any other data or materials generated or used in this study are available from the corresponding authors upon reasonable request. Source data are provided with this paper.

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

## Acknowledgements

We would like to thank Dr. Erik Ingelsson for his support of this project. We also thank the members of Knowles and Svensson labs for valuable discussions and feedback. We would like to thank Dr. Ines Ibañez-Tallon for sharing the *Gpr151* KO mice. We acknowledge the support of the Genetics Bioinformatics Service Center at Stanford University. The study was supported by grants from the National Institutes of Health R01DK120565 (J.W.K.), R01DK116750 (J.W.K.), P30DK116074 (J.W.K., K.J.S.), DK125260 (K.J.S.), DK120565 (K.J.S.), grants from the American Heart Association (J.W.K.), Merck (K.J.S.), the Jacob Churg Foundation (K.J.S.), the McCormick and Gabilan Award (K.J.S.), the Stanford Diabetes Research Center (J.W.K.) and the Stanford Cardiovascular Institute (K.J.S.). Additionally, this study was supported by the American Heart Association (AHA) postdoctoral fellowships (AHA Award Numbers: 18POST34030448 to E.B.M., 905674 to M.Z.), a grant from the Novo Nordisk Foundation and the Stanford Bio-X Program (NNF19OC0054265 to T.M.S.) and the Dean's fellowships at Stanford University (to E.B.M. and P.S.).

## Author contributions

E.B.M. designed the project, performed most experiments, analyzed data and wrote the manuscript with J.W.K. and K.J.S. who both supervised the project; M.Z. performed major experiments; P.H.Z. conducted CLAMS experiments and analyzed the data; J.L., H.K., P.S., C.Y.P., and C.C. contributed experimental work or data analysis; P.N. analyzed RNA-Seq data; T.M.S. queried GWAS summary statistics; A.S. supervised CLAMS experiments; E.B.M., J.W.K., and K.J.S. conceived the project. All authors reviewed the manuscript.

## Competing interests

The authors declare no competing interests.
