## [Peer Review File · Nature Communications]

Reviewers' Comments:

Reviewer #1:

Remarks to the Author:

In this interesting study, Bielczyk-Maczynska and colleagues test the role for hepatic GPR151 in hepatic glucose production and metabolic homeostasis. An inactivating mutation in GPR151 was recently identified in GWAS to be associated with a reduced risk for type 2 diabetes, obesity, and coronary artery disease, and this study follows up on the mechanism that may explain this. Inconsistent with human results – though I do not hold this against the authors! – they find that body weight was not different between WT and GPR151 knockouts; however, GPR151 knockouts exhibited improved glucose and insulin tolerance, reduced cAMP-dependent and gluconeogenic gene expression, reduced pyruvate-induced glucose production, and reduced hepatocyte glucose production. These data suggest that GPR151 may be a key regulator of gluconeogenesis. The data are interesting and the studies generally well performed. While the threshold of requirement for novelty is the decision of the editors, I do have several questions/concerns:

1. Ultimately the take-home message of the paper can be oversimplified as “Downregulation of gluconeogenic enzyme mRNA expression secondary to GPR151 knockdown results in reduced gluconeogenesis.” I do think this is interesting, but considering the large number of settings in which mRNA does not translate to protein does not translate to flux, it is crucial to validate that at least a few of the key proteins of interest are altered in KO mice.
2. The title for Fig 6 is “Overexpression of Gpr151 in the liver of DIO Gpr151 mice rescue the downregulation of gluconeogenesis.” I advise revisiting the phrasing of this, but am also unconvinced that this is an appropriate figure title, because it is not clear to me that the authors show a downregulation of gluconeogenesis in DIO mice.
3. The conclusions regarding the potential translational effects of targeting GPR151 for T2D are overstated. Although it is possible this vein of inquiry may hold promise, considering that the authors did not study a T2D model, they should soften their conclusions in this vein. My personal opinion is that adding a T2D model should not be required (but softening the language should), but of course this will be up to the editor’s discretion.

Reviewer #2:

Remarks to the Author:

Manuscript of Bielczyk-Maczynska et al describes a possible role of orphan receptor GPR151 in glucose metabolism.

The study is original and interesting, the manuscript is well written.

I have just several points which should be addressed:

1. Introduction

- Please, describe peripheral tissues where GPR151 is expressed
- GPR151 KO mice were described previously – it would be great to summarize main findings
- In Results, there is a statement that GPR151 has a known role in regulations of appetite (citation 18), but nothing about this role is said in Introduction – it should be added.

2. Results and Discussion

- Page 6 – insulin and glucagon are not “feeding hormones” (do you mean food intake regulating hormones? These two hormones are not primarily involved in food intake regulation).

3. Methods

- Please, add a source of GPR151 KO mice)
- Design of experiment – scheme is missing, the description of design is quite confusing

Reviewer #3:

Remarks to the Author:

The GPR151 mutation is a rare loss of function mutation that is reported to be protective for both weight gain and type 2 diabetes in humans by some studies. It should be noted that there is conflicting data reported about its biological effect in humans (Gurton and colleagues, PMID 35381001).

The authors show that ablation of the gene results in a reduction in hepatic gluconeogenesis, and a

decrease in the hepatic responsiveness to pyruvate load in mice. Overexpression of the gene in mouse hepatocytes reverses the reduction in hepatic glucose production. This demonstrates a discrete biological effect of the gene that is independent of the reduction in food intake.

They showed that GPR151 KO mice have improved glucose tolerance compared to WT littermates in DIO, without changes in insulin. They show that GPR151 expression in the liver is decreased by feeding. They then injected mice with insulin and reported that this significantly reduced GPR151 expression in SWAT. This may of course be an effect that is mediated by changed in blood glucose, rather than direct hormone action – to demonstrate a direct glucose-independent insulin effect a clamp experiment would be required.

The in vitro work feels partial. They demonstrated that cAMP induction by forskolin increased expression of GPR151, and that dexamethasone did not. It would be satisfying to demonstrate a difference in cellular function between wild type and mutant receptors. I understand the limitation created by an orphan receptor – for instance, in the absence of ligand it is of course not possible to include experiments looking at ligand binding. But it would, for instance, be possible to include experiments looking at cell surface expression to provide a fuller molecular explanation.

They then looked at the transcriptomes of KO mice and compared them to WT littermates. They showed that genes involved in glycolysis and gluconeogenesis were significantly reduced in KO livers. They show that KO mice had a liver-specific reduction in PPARGC1A, a key regulator of energy metabolism. More experiments would be needed to demonstrate if this is the mechanism of action through which it mediates its effects.

They then injected both wild type and KO mice with pyruvate and demonstrated that they had decreased glucose production in KO mice, consistent with a decrease in gluconeogenesis. Figure 4b is described as showing an increase in lactate production, but the p value does not reach significance and therefore this cannot be concluded.

They demonstrated a reduction in glucose secretion using primary cell culture of WT and KO GPR151 hepatocytes in response to glucagon stimulation.

They assessed CREB phosphorylation in response to glucagon in KO vs WT mice and did not demonstrate a significant difference. They did however demonstrate a significant difference in genes regulated by cAMP, with a significant reduction in the KO livers.

Next GPR151 was overexpressed in the livers of DIO GPR151 mice. The level of over-expression was x100 that of the wild type mice. They state that this did not result in a decrease in glucose tolerance. Injection of pyruvate produce the converse picture to the knockout mouse in that pyruvate resulted in a marked increase of glucose production. They refer to figure 5e which does not exist in the manuscript and I presume is a typo.

Overall I think this is a good paper and I find the data on the hepatic role of GPR151 intriguing. They have demonstrated that this is distinct from the previously described role in the habenula, and the protective effect against type 2 diabetes is not simply driven by a reduction in appetite.

Point-by-point response to reviewers' comments

Reviewer #1 (Remarks to the Author):

In this interesting study, Bielczyk-Maczynska and colleagues test the role for hepatic GPR151 in hepatic glucose production and metabolic homeostasis. An inactivating mutation in GPR151 was recently identified in GWAS to be associated with a reduced risk for type 2 diabetes, obesity, and coronary artery disease, and this study follows up on the mechanism that may explain this. Inconsistent with human results – though I do not hold this against the authors! – they find that body weight was not different between WT and GPR151 knockouts; however, GPR151 knockouts exhibited improved glucose and insulin tolerance, reduced cAMP-dependent and gluconeogenic gene expression, reduced pyruvate-induced glucose production, and reduced hepatocyte glucose production. These data suggest that GPR151 may be a key regulator of gluconeogenesis. The data are interesting and the studies generally well performed. While the threshold of requirement for novelty is the decision of the editors, I do have several questions/concerns:

1. Ultimately the take-home message of the paper can be oversimplified as “Downregulation of gluconeogenic enzyme mRNA expression secondary to GPR151 knockdown results in reduced gluconeogenesis.” I do think this is interesting, but considering the large number of settings in which mRNA does not translate to protein does not translate to flux, it is crucial to validate that at least a few of the key proteins of interest are altered in KO mice.

Thank you for your insightful comment. We agree that the lack of gene change validation at the level of protein expression was a limitation of the study. To address this, we have quantified the expression of the rate-limiting hepatic gluconeogenesis enzyme, PEPCK, in the livers of *Gpr151* KO mice compared to WT (Figure 3h,i) and following the overexpression of GPR151 in the livers of *Gpr151* KO mice (Figure 6e,f), and confirmed that the changes in protein expression are consistent with mRNA expression changes and the observed phenotypes. We attach the new figure panels below showing modestly changed protein levels of PEPCK. Despite testing three primary antibodies against G6PC using Western blotting, we found that none of them provided a specific signal to the mouse G6PC protein.

The following sentence has been added in the Results section (page 8): “Further, Western blotting confirmed the modest but significant downregulation of PEPCK in *Gpr151* KO livers at the protein level (Figure 3h,i).”

The following sentence has been added in the Results section (page 11): “In addition, the expression of the rate-limiting hepatic gluconeogenesis enzyme PEPCK in the liver increased significantly in the *Gpr151* KO mice injected with GPR151-expressing AAV compared to controls (Fig. 6e,f).”

Figure 3. h, Representative Western blotting of PEPCK in the livers of 16-week-old DIO *Gpr151* WT and KO male mice. i, Quantification of PEPCK to β -actin in the livers of 16-week-old DIO *Gpr151* WT and KO mice. Two-tailed Student t test. *, $p < 0.05$.

Figure 6. e, Representative Western blotting of PEPCK in the livers of *Gpr151* KO mice with liver-specific GPR151 and GFP overexpression by AAV8. f, Quantification of PEPCK to β -actin in the livers of *Gpr151* KO mice with liver-specific GPR151 and GFP overexpression by AAV8 (N=4, AAV8:GFP; N=4, AAV8:GPR151). Two-tailed Student *t* test; *, $p < 0.05$.

2. The title for Fig 6 is "Overexpression of Gpr151 in the liver of DIO Gpr151 mice rescue the downregulation of gluconeogenesis." I advise revisiting the phrasing of this, but am also unconvinced that this is an appropriate figure title, because it is not clear to me that the authors show a downregulation of gluconeogenesis in DIO mice.

We agree that this figure title overstated our findings. We have changed the Figure 6 title to "Overexpression of *Gpr151* in the liver of DIO *Gpr151* mice rescues the downregulation of hepatic gluconeogenesis gene expression" to better reflect the content of the figure.

3. The conclusions regarding the potential translational effects of targeting GPR151 for T2D are overstated. Although it is possible this vein of inquiry may hold promise, considering that the authors did not study a T2D model, they should soften their conclusions in this vein. My personal opinion is that adding a T2D model should not be required (but softening the language should), but of course this will be up to the editor's discretion.

Thank you for this comment. While we are intrigued about the effects of *Gpr151* loss in T2D models, we aim to explore this separate line of inquiry in our future work. As suggested by the reviewer, we have largely softened our conclusions in this vein and underlined the necessity of additional research to determine if GPR151 function is relevant in the context of T2D, including in the Introduction, Results and Discussion sections of the manuscript.

We have changed the following sentence in the Introduction (page 4): "Our results demonstrate a new function for *Gpr151* in regulating glucose metabolism, suggesting that inhibition of GPR151 may be a novel molecular target for the treatment of T2D." to "Our results demonstrate a new function for *Gpr151* in regulating glucose metabolism."

We have changed the following sentence in the Result section (page 11): "Therefore, due to its novel function in the liver that we have uncovered, GPR151 appears to be a promising target for antidiabetic treatments" to "Therefore, due to its novel function in the liver that we have uncovered, GPR151 appears to be a promising target for pharmacological regulation of blood glucose levels, although its role in relevant models of T2D remains to be studied."

In the Discussion section, we have changed the following sentence (page 13): "The mechanism of hepatic gluconeogenesis regulation by GPR151 should be explored further to understand whether it can be utilized in an orthogonal way to augment currently available therapies for T2D." to "The mechanism of hepatic gluconeogenesis regulation by GPR151 should be explored further to understand whether it is relevant for the control of blood glucose levels in T2D."

Reviewer #2 (Remarks to the Author):

Manuscript of Bielczyk-Maczynska et al describes a possible role of orphan receptor GPR151 in glucose metabolism.

The study is original and interesting, the manuscript is well written.

I have just several points which should be addressed:

1. Introduction

- Please, describe peripheral tissues where GPR151 is expressed
- GPR151 KO mice were described previously – it would be great to summarize main findings
- In Results, there is a statement that GPR151 has a known role in regulations of appetite (citation 18), but nothing about this role is said in Introduction – it should be added.

We appreciate that this reviewer finds our manuscript to be original, interesting and well written.

Gpr151 is widely expressed across many tissues (Regard et al., 2008). We have added the information about all tissues where *Gpr151* is expressed to the Introduction.

The following sentence has been added to the Introduction (page 3): "In addition to the brain, Gpr151 is widely expressed in peripheral tissues in mice, including metabolically relevant tissues such as brown and white adipose tissue, liver, skeletal muscle, and pancreas."

We have added a summary of previous findings on the *Gpr151* KO mice in the Introduction section, including the description of its effect on appetite.

The following sentence has been added in the Introduction section (page 3): "In mice, loss of *Gpr151* results in diminished behavioral responses to nicotine, including less pronounced suppression of appetite".

2. Results and Discussion

- Page 6 – insulin and glucagon are not "feeding hormones" (do you mean food intake regulating hormones? These two hormones are not primarily involved in food intake regulation).

Thank you for pointing out that we misnamed insulin and glucagon as "feeding hormones". We have changed their description to "blood glucose-regulating hormones".

3. Methods

- Please, add a source of GPR151 KO mice)
- Design of experiment – scheme is missing, the description of design is quite confusing

We have added the source of the GPR151 KO mice (page 16). We have also added the design of the main experiment, utilizing GTTs and ITTs in the model of DIO, as new Figure 1a. We attach it below.

Figure 1. a, Schematic of the experiment to determine the effect of *Gpr151* loss on metabolic health in mice.

Reviewer #3 (Remarks to the Author):

The GPR151 mutation is a rare loss of function mutation that is reported to be protective for both weight gain and type 2 diabetes in humans by some studies. It should be noted that there is conflicting data reported about its biological effect in humans (Gurton and colleagues, PMID 35381001).

Thank you for referring this paper, which was published after we submitted our manuscript. We have referenced it in the introduction (page 3).

The authors show that ablation of the gene results in a reduction in hepatic gluconeogenesis, and a decrease in the hepatic responsiveness to pyruvate load in mice. Overexpression of the gene in mouse hepatocytes reverses the reduction in hepatic glucose production. This demonstrates a discreet biological effect of the gene that is independent of the reduction in food intake.

They showed that GPR151 KO mice have improved glucose tolerance compared to WT littermates in DIO, without changes in insulin. They show that GPR151 expression in the liver is decreased by feeding. They then injected mice with insulin and reported that this significantly reduced GPR151 expression in SWAT. This may of course be an effect that is mediated by changed in blood glucose, rather than direct hormone action – to demonstrate a direct glucose-independent insulin effect a clamp experiment would be required.

Thank you for this insightful comment. We agree that the effect of insulin injection on *Gpr151* expression in adipose tissue could be secondary to changes in glucose levels. We added the following sentence in the Discussion section (page 14): "In addition, *Gpr151* expression changes in adipose tissue in response to insulin could be secondary to blood glucose changes. Glucose clamp experiments would be required to discern between the effects of direct insulin action and changes in blood glucose levels."

The in vitro work feels partial. They demonstrated that cAMP induction by forskolin increased expression of GPR151, and that dexamethasone did not. It would be satisfying to demonstrate a difference in cellular function between wild type and mutant receptors. I understand the limitation created by an orphan receptor – for instance, in the absence of ligand it is of course not possible to include experiments looking at ligand binding. But it would, for instance, be possible to include experiments looking at cell surface expression to provide a fuller molecular explanation.

We appreciate this comment by the reviewer, which inspired us to conduct additional studies on the localization and stability of the wild-type and putative loss of function (p.Arg95Ter) GPR151 variants.

We have added the following sentences in the Results section (page 12):

"p.Arg95Ter GPR151 variant results in the loss of the protein

Finally, to determine whether the p.Arg95Ter *GPR151* variant that is associated with lower risk of T2D leads to a functional loss of the GPR151 receptor, we conducted *in vitro* overexpression studies. While wild-type GPR151 receptor was consistently detected in the cellular membrane, the GPR151^{Arg95Ter} variant was undetectable, even though cells were transfected with comparable efficiency (Figure 7, Extended Data Fig. 9). We conclude that the while wild-type GPR151 is localized in cell membrane, the p.Arg95Ter variant likely results in the production of a truncated GPR151 protein that is degraded."

We added these data as Figure 7 and Extended Data Figure 9 in the revised version of the manuscript (attached below).

Figure 7. Cellular localization of the wild-type and p.Arg95Ter GPR151. a, Schematic of the overexpression plasmids used. EGFP and HA-tagged GPR151 are produced in equal ratio from a single transcript. b, Representative confocal images of EGFP-positive cells that were transfected with each of the plasmids, fixed, permeabilized and fluorescently stained against the HA tag. Nuclei counterstained with Hoechst. Images from a single z-plane, 60x objective. Scale bar: 10 μ m. c, Quantification of the ratio between total HA staining (AlexaFluor647) and total GFP signal over the entire cell area in a single confocal z-plane (n=32, *GPR151*^{WT}; n=17, *GPR151*^{Arg95Ter}). Two-tailed Student t test; ****, p<0.0001.

Extended Data Fig. 9 | Supplementary Data to Figure 7. a, Representative confocal images of EGFP-positive cells that were transfected with each of the plasmids, fixed and fluorescently stained against the HA tag without permeabilization. Nuclei counterstained with Hoechst. Images from a single z-plane, 60x objective. Scale bar: 10 μ m. b, Quantification of the ratio between total HA staining (AlexaFluor647) and total GFP signal over the entire cell area in a single confocal z-plane (n=14, *GPR151*^{WT}; n=20, *GPR151*^{Arg95Ter}). Two-tailed Student t test; ****, p<0.0001.

They then looked at the transcriptomes of KO mice and compared them to WT littermates. They showed that genes involved in glycolysis and gluconeogenesis were significantly reduced in KO livers. They show that KO mice had a liver-specific reduction in PPARGC1A, a key regulator of energy metabolism. More experiments would be needed to demonstrate if this is the mechanism of action through which it mediates its effects.

We agree that much more experiments would be needed to determine whether downregulation of PPARGC1A mediates the metabolic effects of *Gpr151* KO. Therefore, we have added the following sentence (page 8): "However, more experiments will be needed to determine whether downregulation of PPARGC1A mediates the metabolic effects of *Gpr151* KO."

They then injected both wild type and KO mice with pyruvate and demonstrated that they had decreased glucose production in KO mice, consistent with a decrease in gluconeogenesis. Figure 4b is described as showing an increase in lactate production, but the p value does not reach significance and therefore this cannot be concluded.

Thank you for pointing that out. We have changed the wording to reflect that the differences in lactate production are not statistically significant. The revised sentence (page 9) states: "Conversely, pyruvate administration resulted in nominal but not statistically significant elevation of blood lactate in *Gpr151* KO mice compared to WT littermates (Figure 4b)".

They demonstrated a reduction in glucose secretion using primary cell culture of WT and KO GPR151 hepatocytes in response to glucagon stimulation.

They assessed CREB phosphorylation in response to glucagon in KO vs WT mice and did not demonstrate a significant difference. They did however demonstrate a significant difference in genes regulated by cAMP, with a significant reduction in the KO livers.

Next GPR151 was overexpressed in the livers of DIO GPR151 mice. The level of over-expression was x100 that of the wild type mice. They state that this did not result in a decrease in glucose tolerance. Injection of pyruvate produce the converse picture to the knockout mouse in that pyruvate resulted in a marked increase of glucose production. They refer to figure 5e which does not exist in the manuscript and I presume is a typo.

Indeed, the reviewer correctly spotted a typo, which we have changed to Figure 4e.

Overall I think this is a good paper and I find the data on the hepatic role of GPR151 intriguing. They have demonstrated that this is distinct from the previously described role in the habenula, and the protective effect against type 2 diabetes is not simply driven by a reduction in appetite.

Reviewers' Comments:

Reviewer #1:

Remarks to the Author:

I believe the authors have responded appropriately to the reviewers' comments and congratulate them on this interesting study.

Reviewer #2:

Remarks to the Author:

I am satisfied with the answers to my question, the manuscript was revised according to my suggestions.

Reviewer #3:

Remarks to the Author:

The GPR151 mutation is a rare loss of function mutation that is reported to be protective for both weight gain and type 2 diabetes in humans by some studies. There is conflicting data on the biological effect in humans, as noted by the authors (Gurton and colleagues, PMID 35381001). The authors show that ablation of the gene results in a reduction in hepatic gluconeogenesis, and a decrease in the hepatic responsiveness to pyruvate load in mice. Overexpression of the gene in mouse hepatocytes reverses the reduction in hepatic glucose production. This demonstrates a discreet biological effect of the gene that is independent of the reduction in food intake. They showed that GPR151 KO mice have improved glucose tolerance compared to WT littermates in DIO, without changes in insulin. They show that GPR151 expression in the liver is decreased by feeding. They then injected mice with insulin and reported that this significantly reduced GPR151 expression in SWAT an effect that is mediated by changed in blood glucose, rather than direct hormone action. As the authors note in the revised manuscript, to demonstrate a direct glucose-independent insulin effect a clamp experiment would be required.

In vitro work with orphan GPCRs is challenging. In the revised manuscript I was satisfied to see that the authors had explored a molecular explanation for the difference in cellular function, by showing a reduction in cell surface expression with the GPR151Arg95Ter variant. They demonstrated that cAMP induction by forskolin increased expression of GPR151, and that dexamethasone did not.

They then looked at the transcriptomes of KO mice and compared them to WT littermates. They showed that genes involved in glycolysis and gluconeogenesis were significantly reduced in KO livers. They show that KO mice had a liver-specific reduction in PPARGC1A, a key regulator of energy metabolism. I agree with the authors that more experiments would be needed to demonstrate if this is the mechanism of action through which it mediates its effects and am satisfied with the manuscript amendment.

They then injected both wild type and KO mice with pyruvate and demonstrated that they had decreased glucose production in KO mice, consistent with a decrease in gluconeogenesis.

They demonstrated a reduction in glucose secretion using primary cell culture of WT and KO GPR151 hepatocytes in response to glucagon stimulation.

They assessed CREB phosphorylation in response to glucagon in KO vs WT mice and did not demonstrate a significant difference. They did however demonstrate a significant difference in genes regulated by cAMP, with a significant reduction in the KO livers.

Next GPR151 was overexpressed in the livers of DIO GPR151 mice. The level of over-expression was x100 that of the wild type mice. They state that this did not result in a decrease in glucose tolerance. Injection of pyruvate produce the converse picture to the knockout mouse in that pyruvate resulted in a marked increase of glucose production.

Overall I think this is a good paper and I find the data on the hepatic role of GPR151 intriguing. They have demonstrated that this is distinct from the previously described role in the habenula, and the protective effect against type 2 diabetes is not simply driven by a reduction in appetite. I

am satisfied with the alterations they have made to the script, including the extra data provided, and recommend this manuscript for publication.

Point-by-point response to reviewers' comments

Reviewer #1 (Remarks to the Author):

I believe the authors have responded appropriately to the reviewers' comments and congratulate them on this interesting study.

We thank the reviewer for the insightful comments and are happy that we have managed to address them satisfactorily.

Reviewer #2 (Remarks to the Author):

I am satisfied with the answers to my question, the manuscript was revised according to my suggestions.

We thank the reviewer for the insightful comments and are happy that we have managed to address them satisfactorily.

Reviewer #3 (Remarks to the Author):

The GPR151 mutation is a rare loss of function mutation that is reported to be protective for both weight gain and type 2 diabetes in humans by some studies. There is conflicting data on the biological effect in humans, as noted by the authors (Gurton and colleagues, PMID 35381001).

The authors show that ablation of the gene results in a reduction in hepatic gluconeogenesis, and a decrease in the hepatic responsiveness to pyruvate load in mice. Overexpression of the gene in mouse hepatocytes reverses the reduction in hepatic glucose production. This demonstrates a discreet biological effect of the gene that is independent of the reduction in food intake.

They showed that GPR151 KO mice have improved glucose tolerance compared to WT littermates in DIO, without changes in insulin. They show that GPR151 expression in the liver is decreased by feeding. They then injected mice with insulin and reported that this significantly reduced GPR151 expression in SWAT an effect that is mediated by changed in blood glucose, rather than direct hormone action. As the authors note in the revised manuscript, to demonstrate a direct glucose-independent insulin effect a clamp experiment would be required.

In vitro work with orphan GPCRs is challenging. In the revised manuscript I was satisfied to see that the authors had explored a molecular explanation for the difference in cellular function, by showing a reduction in cell surface expression with the GPR151Arg95Ter variant. They demonstrated that cAMP induction by forskolin increased expression of GPR151, and that dexamethasone did not.

They then looked at the transcriptomes of KO mice and compared them to WT littermates. They showed that genes involved in glycolysis and gluconeogenesis were significantly reduced in KO livers. They show that KO mice had a liver-specific reduction in PPARGC1A, a key regulator of energy metabolism. I agree with the authors that more experiments would be needed to demonstrate if this is the mechanism of action through which it mediates its effects and am

satisfied with the manuscript amendment.

They then injected both wild type and KO mice with pyruvate and demonstrated that they had decreased glucose production in KO mice, consistent with a decrease in gluconeogenesis.

They demonstrated a reduction in glucose secretion using primary cell culture of WT and KO GPR151 hepatocytes in response to glucagon stimulation.

They assessed CREB phosphorylation in response to glucagon in KO vs WT mice and did not demonstrate a significant difference. They did however demonstrate a significant difference in genes regulated by cAMP, with a significant reduction in the KO livers.

Next GPR151 was overexpressed in the livers of DIO GPR151 mice. The level of over-expression was x100 that of the wild type mice. They state that this did not result in a decrease in glucose tolerance. Injection of pyruvate produce the converse picture to the knockout mouse in that pyruvate resulted in a marked increase of glucose production.

Overall I think this is a good paper and I find the data on the hepatic role of GPR151 intriguing. They have demonstrated that this is distinct from the previously described role in the habenula, and the protective effect against type 2 diabetes is not simply driven by a reduction in appetite. I am satisfied with the alterations they have made to the script, including the extra data provided, and recommend this manuscript for publication.

We thank the reviewer for the insightful comments and are happy that we have managed to address them satisfactorily.